# Adaptive Quality-Diversity Trade-offs for Large-Scale Batch Recommendation

## Abstract

A core research question in recommender systems is to propose batches of highly relevant and diverse items, that is, items personalized to the user's preferences, but which also might get the user out of their comfort zone. This diversity might induce properties of serendipity and novelty which might increase user engagement or revenue. However, many real-life problems arise in that case: e.g., avoiding to recommend distinct but too similar items to reduce the churn risk, and computational cost for large item libraries, up to millions of items. First, we consider the case when the user feedback model is perfectly observed and known in advance, and introduce an efficient algorithm called B-DivRec combining determinantal point processes and a fuzzy denuding procedure to adjust the degree of item diversity. This helps enforcing a quality-diversity trade-off throughout the user history. Second, we propose an approach to adaptively tailor the quality-diversity trade-off to the user, so that diversity in recommendations can be enhanced if it leads to positive feedback, and vice-versa. Finally, we illustrate the performance and versatility of B-DivRec in the two settings on synthetic and real-life data sets on movie recommendation and drug repurposing.

## 1 Introduction

Preserving user engagement, that is, the willingness of users to query a recommender system and to interact with recommended items, is crucial and yet a difficult task. It is well-known that, beyond recommending merely the most popular items or those closest to the estimated user's interests, introducing diversity in recommendation is key to avoid the churn risk, *i.e.*, customer attrition (Poulain & Tarissan, 2020). This topic has been widely studied under the name of "diverse/novel recommendations" or "serendipity" (Abbassi et al., 2009; Kotkov et al., 2018; Ziarani & Ravanmehr, 2021). The rationale behind it appears in several real-life contexts: for instance, diversity might increase revenue by keeping user engagement high in the movie streaming or music industry (Van den Oord et al., 2013; Anderson et al., 2020); get a teenager out of their comfort zone and make them discover new cultural goods (Ibrahim et al., 2025); evaluate the global state of a student's knowledge on a specific subject in education (Chavarriaga et al., 2014; Yanes et al., 2020); or discover a first-in-class drug treatment in the pharmaceutical industry, where it has been shown that first-in-class molecules might generate higher revenue compared to well-known classes of molecules with a therapeutic advantage ("best-in-class") (Schulze & Ringel, 2013; Spring et al., 2023). All in all, the goal in recommender systems is to satisfy apparently contradictory objectives: to recommend user-personalized items which introduce diversity in the user's history of recommended items.

Moreover, when implementing a recommender system for real-life applications, one must also face the problems of developing computationally tractable pipelines on large libraries of items (Cha et al., 2018). Last but not least, the definition of diversity and quantifying the diversity in the recommended (batch of) items is tricky by itself, as illustrated by the multiple definitions used in the literature (refer to Appendix A for an overview). Diversity might be understood in terms of "intrabatch" (local) or "interbatch" (global) (Bederina & Vie, 2025), respectively meaning that the batch of recommended items at a specific round should be diverse, or the batch of recommended items should be diverse with respect to the user's prior history of selected items. This paper also tackles a novel problem of the literature, that is, adaptively tuning the level of diversity in the recommendations to the user. The tolerance to diversity might indeed vary from one user to the other, and contributes to the user engagement in the recommender system (Xu & Matsumura, 2024). No other paper to the best of our

knowledge has attempted to propose an automated procedure for tuning the level of diversity at an early stage of the recommendation pipeline.

**Problem setting.** We denote $N$ the total number of items (potentially of the order of millions) in the universe $\Omega := \{1, 2, \ldots, N\}$. Recommendation rounds happen sequentially: (1) a (possibly new) user $\boldsymbol{h}^t$ queries the recommender system at each time $t$, (2) the recommender system returns a fixed-size batch $\mathcal{S}^t \subset \Omega$ of $B$ items to present to the user, (3) user $\boldsymbol{h}^t$ outputs feedback values $\boldsymbol{y}^t := \{y_1^t, \ldots, y_B^t\}$ for each recommendation. We choose to ignore the specific identity of items and users, and instead, respectively define them with the item embedding $\boldsymbol{\phi}^i \in \mathbb{R}^d$ for item $i$, or by the *summary* of user history $\boldsymbol{h}^t \in \mathbb{R}^m$ at each recommendation round $t$. Ideally, $\boldsymbol{h}^t$ captures all the information about the items the user has *previously interacted with* (*e.g.*, $\boldsymbol{h}^t$ can describe how much one of the $m$ categories is liked). It can be hard-coded (one-hot encoding of liked item categories), or an embedding learned over previously collected items, or the result of some dimension reduction algorithm applied to the embeddings of positively interacted with items. We define $\mathcal{H}^t$ the user history of prior recommendations to user $\boldsymbol{h}^t$ up to the round $t$ (not included). To define the similarity between items in a flexible fashion, we select a kernel function $k : \mathbb{R}^d \times \mathbb{R}^d \mapsto \mathbb{R}$. $I_n$ is the identity matrix of size $n$. Finally, to make good and *personalized* recommendations–regardless of diversity– we aim to learn the feedback model $q_\Theta : \mathbb{R}^d \times \mathbb{R}^m \to \mathbb{R}$ where $q_\Theta(\boldsymbol{\phi}, \boldsymbol{h})$ is the expected feedback for user $\boldsymbol{h}$ on item $\boldsymbol{\phi}$ which should be maximized. In the case of a known and noiseless feedback model, if $\boldsymbol{\phi}$ is the $k^{\text{th}}$ recommended item to user $\boldsymbol{h}^t$ at time $t$, then $y_k^t = q_\Theta(\boldsymbol{\phi}, \boldsymbol{h}^t)$.

To quantify the quality and the diversity of recommendations, we define the following pointwise metrics for any round $t > 0$. We are aware that an abundant literature on both quality and diversity in recommender systems exist, see Appendix A where they are discussed. We decided to separately assess quality and diversity. Quality (also called relevance in this paper and denoted rel) is simply defined by the expected click-through rate. We also consider the precision metric (denoted prec), that is, the ratio of positively rated items (*i.e.*, such that the feedback value is higher than a given threshold $\tau > 0$) over the total number of recommended items. Moreover, we define two types of diversity: the *intrabatch/local diversity* (denoted $\text{div}^{\text{L}}$), focusing on the diversity inside a batch of $B$ items; and the *interbatch/global diversity* (denoted $\text{div}^{\text{G}}$), which looks at the diversity of the previously and currently recommended items, meaning that it also takes into account the user history. An intuitive idea of how diverse a set $\mathcal{S}$ of items is can be obtained by computing the volume of the parallelotope induced by the rows of the kernel matrix $K_{\mathcal{S},\mathcal{S}} := (k(\boldsymbol{\phi}^i, \boldsymbol{\phi}^j))_{i,j \in \mathcal{S}}$ built from the item embeddings in $\mathcal{S}$. The volume of a set $\mathcal{S} := \{i_1, \ldots, i_B\}$ of items is $\text{vol}(\mathcal{S}) := (\det K_{\mathcal{S},\mathcal{S}})^{1/2}$. [1]

$$\forall t > 0, \ \text{rel}(\mathcal{S}^t) := \frac{1}{B} \sum_{i \in \mathcal{S}^t} q_\Theta(\boldsymbol{\phi}^i, \boldsymbol{h}^t), \quad \text{div}^{\text{L}}(\mathcal{S}^t) := \text{vol}(\mathcal{S}^t), \quad \text{div}^{\text{G}}(\mathcal{S}^t) := \text{vol}(\mathcal{S}^t \cup \mathcal{H}^t). \quad (1)$$

To assess the performance accrued by a strategy across $T$ recommendation rounds, we consider the average of rel, prec, $\text{div}^{\text{L}}$ and $\text{div}^{\text{G}}$ across rounds. Moreover, we also consider a summary metric which mixes quality and diversity at the end of $T$ recommendation rounds. Given a threshold $\tau$, the effective diversity $\text{div}^{+}$ contributed by positively-labeled items for a single user $\boldsymbol{h}$ after $T$ rounds is

$$\text{div}^{+}(\{\mathcal{S}^1, \ldots, \mathcal{S}^T\}, \boldsymbol{h}) := \text{vol}(\{i \mid i \in \bigcup_{s < t} \mathcal{S}^s, q_\Theta(\boldsymbol{\phi}^i, \boldsymbol{h}) \geq \tau\}). \quad (2)$$

The final summary metric is the average $\text{div}^{+}$ across encountered users.

**Contributions.** Our objective is to find a principled and scalable approach to implement the quality-diversity trade-off–in the large sense: any kernel, any feedback model, or as general as possible–for a possibly large number of items up to millions. It means that anything beyond a time complexity linear in the number of items $N$ will be intractable. First, we introduce an efficient algorithm called B-DivRec combining determinantal point processes and a fuzzy denuding procedure to adjust the degree of item diversity. This helps enforcing a quality-diversity trade-off throughout the user history when the user feedback model is known (Section 3). Second, we propose an approach to adaptively tailor the quality-diversity trade-off to the user, so that diversity in recommendations can be enhanced if it leads to positive feedback, and vice-versa (Section 4). Finally, we illustrate the performance and versatility of B-DivRec in the two settings on synthetic and real-life data sets on

---

[1]By definition of a kernel function, this definition of volume is well-defined for any $\mathcal{S} \subseteq \Omega$.

movie recommendation and drug repurposing (Section 5). We conclude the paper by discussing the potential and the limitations of B-DivRec in Section 6.

## 2 RELATED WORKS

As mentioned in the introduction, the topics addressed in this paper are related to a large section of the literature on recommendation (serendipity, avoidance of the churn risk, tractable recommender systems). Due to the space limit, we refer the reader to Appendix A for a more comprehensive review of the literature. We focus here on fundamental notions in DPPs and baselines in Section 5.

**Determinantal Point Processes.** A point process is a distribution over finite subsets of a (finite) set $\Omega$. A determinantal point process (DPP) is a process where the probability of sampling a subset $A$ is correlated to the determinant of the kernel function $k$ applied to this subset, that is, $\mathbb{P}(\mathcal{S} = A) \propto \det k_{A,A}$ (Macchi, 1975). Here, we consider the definition of DPPs with $L$-ensembles, as described in Borodin & Rains (2005), and $k_{\Omega,\Omega}$ is a positive semi-definite matrix. Intuitively, the higher the volume formed by the item embeddings in this subset, the higher the probability. Sampling algorithms for DPPs are naively in $\mathcal{O}(N^3)$ (Hough et al., 2006, Theorem 7 and Algorithm 18), but their complexity may be reduced down to $\mathcal{O}(\alpha N \cdot \mathrm{poly}(B))$ when sampling a subset of fixed size $B$ with an $\alpha$-DPP (Calandriello et al., 2020) where $\alpha \leq 1$. However, one might want to find the subset with highest probability–that is, the most diverse–instead of sampling according to the DPP. The associated maximization problem is NP-hard (Ko et al., 1995; Grigorescu et al., 2022), but there are greedy approximations in $\mathcal{O}(B^2 N)$ (Gillenwater et al., 2012; Chen et al., 2018) for maximizing over subsets of size $B$. Finally, conditioning over another subset $\mathcal{H}$ of items– that is, sampling a subset which is diverse compared to previously selected set of points–can be described with the following distribution (Borodin & Rains, 2005) $\mathbb{P}(\mathcal{S} = A \cup \mathcal{H} \mid \mathcal{S} \supseteq \mathcal{H}) \propto \det\left(k_{A,A} - k_{A,\mathcal{H}} k_{\mathcal{H},\mathcal{H}}^{-1} k_{A,\mathcal{H}}^{\mathsf{T}}\right)$. This is a simple approach to integrating the user history to the recommendation. However, conditioning has a dependency in $\Omega(|\mathcal{H}|^3)$ where $\mathcal{H}$ is the user history, and inversion of the history-related kernel matrix might be expensive. [2]

To tackle the issue of recommending items with high quality/relevance and high "intrabatch"/local diversity, Kulesza & Taskar (2010) introduced the quality-diversity (QD) decomposition of a DPP. Given $N$ item embeddings $\{\phi^i\}_{i \in \Omega}$ such that $\|\phi^i\|_2 = 1$ for any $i \in \Omega$, and positive quality scores $\{q_\Theta(\phi^i, \boldsymbol{h})\}_{i \in \Omega}$ for each item and a given user $\boldsymbol{h}$, the QD decomposition is the distribution $\mathbb{P}(\mathcal{S} = A) \propto \det(Q_A \Phi_A \Phi_A^{\mathsf{T}} Q_A)$, where $Q_A$ is a diagonal matrix of size $B \times B$ which contains the quality scores for each item $i_1, \ldots, i_B$ in $A$, and $\Phi_A := [\phi^{i_1}, \ldots, \phi^{i_B}]^{\mathsf{T}}$ is the row-concatenation of all item embeddings in $A$. Many papers relied on this approach (Gong et al., 2014; Wilhelm et al., 2018; Zhan et al., 2021; Svensson et al., 2025; Xuan et al., 2025). However, this decomposition might be a bit restrictive, as the control of the quality-diversity trade-off is limited, and only linear or RBF (Affandi et al., 2014; Wilhelm et al., 2018) kernels are considered. Affandi et al. (2012) is closer to our objective of increasing the "interbatch" or global diversity across consecutive recommendation rounds, and introduces Markov DPPs applied to the daily recommendation of news headlines. The main idea is to condition the subset of items sampled at round $t + 1$ on the subset sampled at round $t$. Their construction guarantees that the DPP marginals are maintained. Finally, the authors also study in their experiments the unknown feedback model setting, and tackle this problem by updating empirical quality scores for each item over time in the QD decomposition. However, their procedure is quite costly and not tractable in large data sets (see our experiments in Section 5).

**Non-DPP recommender systems for diversity.** Besides DPPs, there is a plethora of other approaches, including Maximal Marginal Relevance (MMR) (Carbonell & Goldstein, 1998). The problem originally tackled by MMR is slightly different from ours: given a library of items $\Omega$, an item-item similarity function sim, a relevance function rel between items, and a quality-diversity trade-off parameter $\lambda \in [0, 1]$, the goal is to return an item $\phi^t$ at round $t$ which is both relevant to the query item $\boldsymbol{h}^{\mathrm{query}}$ and diverse with respect to previously selected items $i_1, i_2, \ldots, i_{t-1}$. The recommendation procedure is defined recursively. At time $t$, MMR se-

---

[2]We abuse notation here, as it is clear that we refer to the complexity notation $\Omega$ and not the universe of items $\Omega$.

lects item $i_t := \arg\max_{i \in \Omega} \left( \lambda \, \mathrm{rel}(\phi^i, h^{\mathrm{query}}) - (1 - \lambda) \max_{j \in \{i_1, \dots, i_{t-1}\}} \mathrm{sim}(\phi^i, \phi^j) \right)$ for each $t \in \{1, \dots, B\}$, where the maximum value over $\emptyset$ is set to 0 by convention. However, it is easy to rewrite the expression in MMR to fit the setting where a user context and history is provided as query. Instead of a relevance function, we consider the feedback model and the kernel function as a similarity metric. Using notation from above, with $b \in \{1, \dots, B\}$, at time $t$, MMR selects item $i^{t+b} := \arg\max_{i \in \Omega} \left( \lambda \, q_\Theta(\phi^i, h^t) - (1 - \lambda) \max_{j \in \mathcal{H}^t \cup \{i^t, \dots, i^{t+b-1}\}} k(\phi^i, \phi^j) \right)$. Yet, it would be interesting to strengthen even further the constraint on diversity, and to leverage the literature on DPPs to sample a whole diverse subset instead of an iterative, greedy procedure.

## 3 A GENERIC APPROACH FOR THE QUALITY-DIVERSITY TRADE-OFF IN DPPs

In this section, we define our trade-off on relevance and diversity, instead of a multi-objective task (Abbassi et al., 2009) or a multi-step process for filtering relevance or diversity (Ibrahim et al., 2025; Yuan et al., 2016). We recall that $k : \mathbb{R}^d \times \mathbb{R}^d \to \mathbb{R}$ is a similarity kernel on items. To use a quality-diversity decomposition, we need:

**Assumption 3.1.** Positive-definite kernel. *The kernel $k$ is a* positive-definite kernel, *that is, there exists $d' \in \mathbb{N}^*$ and a feature map function $\nu : \mathbb{R}^d \to \mathbb{R}^{d'}$ such that, for any $x, y \in \Omega$, $k(x, y) = \nu(x)^\intercal \nu(y)$. $\nu$ is extended to subsets of items: $\nu(\mathcal{S}) := (\nu(\phi^i))_{i \in \mathcal{S}} \in \mathbb{R}^{B \times d'}$ for any $\mathcal{S} \subseteq \Omega$.*

**Assumption 3.2.** Positive unbounded feedback. *The feedback model $q_\Theta$ has values in $\mathbb{R}_+^*$.*

**Assumption 3.3.** Unit embeddings. *$\|\phi\|_2 = \|h\|_2 = 1$ for all user embeddings $h \in \mathbb{R}^m$ and item embeddings $\phi \in \Omega$.*

Note that those assumptions are only moderately restrictive, as adequate kernel functions (linear, RBF, Matérn (Williams & Rasmussen, 2006; Duvenaud, 2014)) and classification or regression models with positive values (Lee & Seung, 1999; Wood, 2017) abound in the literature, and renormalizing embeddings is a common procedure. Then, we assume that the feedback model is known and perfect, and that the observed user feedback is noiseless.

**Assumption 3.4.** Noiseless observed feedback. *At a given recommendation round $t > 0$ to user $h^t$, if $\phi$ is the $k^{th}$ recommended item, then the observed feedback value $y_k^t$ from user $h^t$ for the $k^{th}$ item satisfies $y_k^t = q_\Theta(\phi, h^t)$.*

**The Quality-Diversity Trade-off (QDT) family.** Under these assumptions, we extend the quality-diversity decomposition (Kulesza & Taskar, 2010) in determinantal point processes to any valid kernel, and we allow to interpolate from only quality-focused recommendation [3] to diversity-focused recommendation [4]. The likelihood matrix for this family of DPPs, called QDT, for a given user $h$, their history $\mathcal{H}$, and item subset $\mathcal{S}$, is:

$$\forall \mathcal{S} \subseteq \Omega, \forall h \in \mathbb{R}^m, \; L_f^\lambda(\mathcal{S}; h) := (Q_{h,\mathcal{S}})^{2\lambda} f(k, \mathcal{S}, \mathcal{H})^{2(1-\lambda)} (Q_{h,\mathcal{S}})^{2\lambda}, \tag{3}$$

where $\lambda \in [0, 1]$ controls the trade-off between quality ($\lambda = 1$) and diversity ($\lambda = 0$). The presence of $\lambda$ allows us to implement continuously and explicitly the trade-off between relevance and diversity. $Q_{h,\mathcal{S}}$ is a diagonal matrix containing the expected rewards $\{q_\Theta(\phi^i, h)\}$ for $i \in \mathcal{S}$. $f$ is a function with values in the set of positive-definite matrices in $\mathbb{R}^{B \times B}$, and depends on kernels computed on $S, \Omega, \mathcal{H}$. $f$ should incorporate all information about the desired diversity in recommended items. Note that we completely disentangle quality and diversity in this definition.

Equation 3 offers a flexible definition of the quality-diversity trade-off, which recovers various known DPPs described in Section 2 by defining $f$ and $\lambda$. For instance, traditional quality-diversity decomposition (Kulesza & Taskar, 2010) can be obtained by setting $f^{\mathrm{QDDecomp}}(k, \mathcal{S}, \mathcal{H}) := k_{\mathcal{S},\mathcal{S}}$ with $\lambda = 0.5$, where $k$ is the linear kernel $k(\{\psi\}, \{\phi\}) = \psi^\intercal \phi$ or, equivalently, $\nu : x \mapsto x$. A conditional DPP (Borodin & Rains, 2005) can be described using $\lambda = 0.5$ and $f^{\mathrm{CondDPP}}(k, \mathcal{S}, \mathcal{H}) := k_{\mathcal{S},\mathcal{S}} - k_{\mathcal{S},\mathcal{H}}(k_{\mathcal{H},\mathcal{H}})^{-1}(k_{\mathcal{S},\mathcal{H}})^\intercal$. Finally, an $\varepsilon$-greedy approach can be obtained by setting $f^{\varepsilon\text{-greedy}}(k, \mathcal{S}, \mathcal{H}) := \mathrm{I}_B$ with $\lambda = 0.5$ $\varepsilon\%$ of the rounds (greedy phase), and setting

---

[3] That is, we do not take into account the diversity described by the kernel.

[4] *i.e.*, we ignore quality scores and only aim at being diverse

---

**Algorithm 1** Recommendation round at $t > 0$ with B-DivRec and an adaptive Q-D trade-off.

---

1: **Input** $\boldsymbol{h}^t$: user context, $\mathcal{H}^t$: user history, $q_\Theta$: feedback model, $\lambda^t$: initial relevance weight
2: **Parameters** $k$: kernel function, $f = f^{\text{B-DivRec}}$, $\mathcal{A}$: online learner
3: **Output** $\mathcal{S}^t$: $B$ item recommendations, $\lambda^{t+1}$: updated relevance weight
4: $Q_{\boldsymbol{h}^t,\Omega} \leftarrow \text{diag}(\{q_\Theta(\boldsymbol{\phi},\boldsymbol{h}^t)\}_{\boldsymbol{\phi}\in\Omega})$ and initialize $\mathcal{A}$ with $\lambda^t$
5: $L_{f^{\text{B-DivRec}}}^{\lambda^t}(\Omega;\boldsymbol{h}^t) \leftarrow (Q_{\boldsymbol{h}^t,\Omega})^{2\lambda^t} f(k,\Omega,\mathcal{H}^t)^{2(1-\lambda^t)}(Q_{\boldsymbol{h}^t,\Omega})^{2\lambda^t}$    # Likelihood matrix
6: **Use SAMPLING** Sample $\mathcal{S}^t \subseteq \Omega$ proportionally to $\det L_f^{\lambda^t}(\mathcal{S}^t;\boldsymbol{h}^t)$
7: **or MAXIMIZATION** Solve $\mathcal{S}^t \in \arg\max_{\mathcal{S}\subseteq\Omega} \det L_f^{\lambda^t}(\mathcal{S};\boldsymbol{h}^t)$
8: $\boldsymbol{y}^t \leftarrow (y_1^t, y_2^t, \ldots, y_B^t)$, update $\mathcal{A}$ with $(1-2\lambda^t)\nabla_\lambda \text{scr}_{f,\boldsymbol{y}^t}^{\lambda^t}(\mathcal{S}^t;\boldsymbol{h}^t)$ and select $\lambda^{t+1}$ from $\mathcal{A}$
9: **return** $\mathcal{S}^t, \lambda^{t+1}$

---

$f^{\varepsilon\text{-greedy}}(k,\mathcal{S},\mathcal{H}) := k_{\mathcal{S},\mathcal{S}}$ with $\lambda = 0$ the other $(1-\varepsilon)\%$ of rounds (exploratory phase). We now consider the log-determinant of $L_f^\lambda(\mathcal{S};\boldsymbol{h})$ as the score scr of any $B$-sized set $\mathcal{S}$

$$\forall \mathcal{S} \subseteq \Omega, \forall \boldsymbol{h} \in \mathbb{R}^m, \; \text{scr}_{f,q_\Theta}^\lambda(\mathcal{S};\boldsymbol{h}) := 4(1-\lambda)\log \text{vol}(f(k,\mathcal{S},\mathcal{H})) + 4\lambda \sum_{\boldsymbol{\phi}\in\mathcal{S}} \log q_\Theta(\boldsymbol{\phi},\boldsymbol{h}), \quad (4)$$

with the convention $\text{scr}_{f,q_\Theta}^\lambda(\emptyset) := 0$, provided that the volume is well-defined–that is, $f(k,\Omega,\emptyset)$ is positive-definite for any universe of items $\Omega$. From the expression in Equation 4, the larger the score, the better the subset $\mathcal{S}$ for the quality-diversity trade-off. We could stop at this point, and simply consider a (possibly conditional) DPP to obtain diversified recommendations (Affandi et al., 2012). Still, as discussed in the introduction, this approach might be expensive for users with long histories. We suggest another, more tractable, approach named B-DivRec below, which leverages the existence of kernel-associated feature maps (Assumption 3.1).

**The B-DivRec DPP.** B-DivRec belongs to the QDT family, and is defined with $\alpha \in [0,2]$

$$f^{\text{B-DivRec}}(k,\mathcal{S},\mathcal{H};\alpha) := \left(\nu(\mathcal{S}) - \nu(g(\mathcal{S};\mathcal{H},\alpha))\right)\left(\nu(\mathcal{S}) - \nu(g(\mathcal{S};\mathcal{H},\alpha))\right)^\mathsf{T} \text{ where} \quad (5)$$

$$\forall i \leq B, \; g(\mathcal{S};\mathcal{H},\alpha)_{i,\cdot} := \begin{cases} \boldsymbol{\phi}^{\ell_i} & \text{if } \max_{j\in\mathcal{H}} \cos(\nu(\boldsymbol{\phi}^{\ell_i}),\nu(\boldsymbol{\phi}^j)) \geq 1-\alpha \\ \mathbf{0} & \text{otherwise} \end{cases} \in \mathbb{R}^{B\times d} \text{ if } \mathcal{S}^: = \{\ell_i\}_i,$$

and $\cos(x,y)$ is the cosine similarity between vectors $x$ and $y$. $\alpha$ controls in a more subtle way than the kernel $k$ the degree of diversity expected compared to the user history, by filtering out items too similar to the history. We discuss the choice of hyperparameter $\alpha$ in Section 5. See Algorithm 1 for a pseudo-code of B-DivRec with a fixed value of $\lambda \in [0,1]$. (Gartrell et al., 2017; Dupuy & Bach, 2018) also propose a family of low-rank factorizations of $L$-matrices. However, Gartrell et al. (2017) does not endorse a quality-diversity trade-off, whereas Dupuy & Bach (2018) requires two supplementary low-rank approximations. Conversely, B-DivRec straightforwardly uses the Nyström approximation to obtain computations in $\mathbb{R}^{d'}$ where $d' \ll d$.

**Efficiency of B-DivRec.** Naively, the computation of the $L$-matrix in B-DivRec is in $\mathcal{O}(N^3)$. However, we leverage several approximations and methods to achieve a time complexity linear in $N$ that we describe more thoroughly in Appendix B. In practice, we learn function $\nu$ with a Nyström approximation (Nyström, 1930; Yang et al., 2012; Liu et al., 2021) of low rank $d'$ to avoid computations on potentially large matrices of size $N \times N$. This approximation can be applied to any kernel function $k$. However, one should be careful to select $d'$ such that $N \gg d' \geq B + \max_{t\leq T} |\mathcal{H}^t|$ to ensure that the interbatch/global volume of any sampled batch of $B$ items can be larger than $0$ (see metrics in Section 1). The Nyström approximation of rank $d'$ on a random selection of representative points, run once, has a time complexity of $\mathcal{O}(N(d')^2 + (d')^3)$ (Williams & Seeger, 2000). The computation of $g(\mathcal{S};\mathcal{H},\alpha)$ for any user $\boldsymbol{h}$ of history $\mathcal{H}$ and subset $\mathcal{S}$ using a $k$-d tree leads to an average time complexity in $\Omega(B\log|\mathcal{H}|)$, where $|\mathcal{H}| \ll N$ as a general rule, and $\Omega(B|\mathcal{H}|)$ in the worst case of unbalanced trees (Arya & Mount, 1993). Retrieving the closest neighbor can also be computed on large sets of (feature maps of) items by considering an approximate nearest neighbor algorithm, *e.g.*, FAISS (Douze et al., 2024) or LSH (Dasgupta et al.,

2011). The computation of the matrix power (in $\lambda$) can also be in linear time in $N$. Using the $\alpha$-DPP sampling procedure (Calandriello et al., 2020) for the SAMPLING strategy and the greedy algorithm in (Chen et al., 2018) for the MAXIMIZATION approach, with time complexities linear in $N$ as described in Section 1, we confirm that B-DivRec remains tractable even when faced with millions of items, as demonstrated in the experiments. Moreover, we delve into supplementary details of implementation in Appendix B.

## 4 ADAPTIVE QUALITY-DIVERSITY (Q-D) TRADE-OFF

A frequent problem when dealing with the quality-diversity trade-off is to handle the fact that some users might be more receptive to diverse recommendations than others. We wish for an automated procedure for tuning the trade-off–in practice, changing the value of hyperparameter $\lambda$–which takes into account the user's prior reactions to diversified items. We can frame this problem as an online learning game between a player–that is, the recommender system–and Nature–which is the interaction with *one* user $(\boldsymbol{h}, \mathcal{H}^t)$ querying the recommender system at time $t$ (Auer et al., 2002; De Rooij et al., 2014). At round $t \leq T$ of the game, the recommender system chooses a value of $\lambda^t \in [0, 1]$. Then, the recommender system makes a recommendation $\mathcal{S}^t$ to user $\boldsymbol{h}$ and receives a loss (or gain (De Rooij et al., 2014)) value related to the quality-diversity trade-off achieved, depending on the vector of feedback $\boldsymbol{y}^t$ from Nature. The recommender system should use this loss to update and use $\lambda^{t+1} \in [0, 1]$ at the next round. The game ends after $T$ interactions with the same user. The quality-diversity trade-off gain is given by the function $\lambda, \mathcal{S}, \boldsymbol{y} \mapsto \mathrm{scr}^\lambda_{f\text{B-DivRec}, \boldsymbol{y}}(\mathcal{S}; \boldsymbol{h})$ for all $\lambda \in [0, 1]$, $\mathcal{S} \subseteq \Omega$, $|\mathcal{S}| = B$, and $\boldsymbol{y} \in (\mathbb{R}^*_+)^B$. [5] The goal of the game, by selecting the $\lambda_t$'s, is to maximize the cumulated quality-diversity trade-off, and alternatively, to minimize the cumulative regret compared to a deterministic oracle which knows in advance the pairs batches-feedbacks $(\mathcal{S}^t, \boldsymbol{y}^t)_{t \leq T}$:

$$\mathcal{R}^{\text{adapt}}(T; \boldsymbol{h}) := \max_{\lambda \in [0,1]} \sum_{t \leq T} \mathrm{scr}^\lambda_{f\text{B-DivRec}, \boldsymbol{y}^t}(\mathcal{S}^t; \boldsymbol{h}) - \mathrm{scr}^{\lambda^t}_{f\text{B-DivRec}, \boldsymbol{y}^t}(\mathcal{S}^t; \boldsymbol{h}) . \tag{6}$$

Many online learners have been introduced to solve this type of problem, *e.g.*, EXP3 (Auer et al., 2002) and AdaHedge (De Rooij et al., 2014). We select AdaHedge, as it is an online learner which is horizon $T$-agnostic, and does not require to know the scale of the gain function in advance. We further go into details as regards the implementation of this procedure in Appendix C.1. Algorithm 1 (purple lines) shows how we modify the initial recommendation algorithm with B-DivRec to adaptively choose $\lambda$. We derive guarantees on the regret incurred by the adaptive learning procedure.

**Theorem 4.1.** Upper bound on the regret incurred by the adaptive diversity tuning procedure. *An upper bound on the regret $\mathcal{R}^{adapt}(T; \boldsymbol{h})$ incurred by the adaptive strategy for tuning the level of diversity $\lambda \in [0, 1]$ for user $\boldsymbol{h}$ over $T$ rounds of recommendations is*

$$\mathcal{R}^{adapt}(T; \boldsymbol{h}) \leq 2\delta_T \sqrt{T \log(2)} + 16\delta_T(2 + \log(2)/3), \ where \ \delta_T := 8 \max_{t \leq T} \log \frac{(\max_{i \leq B} y_i^t)^B}{vol(f(k, S^t, H^t))} .$$

The proof is shown in Appendix C.2. The upper bound has a time dependence in $\mathcal{O}(\sqrt{T})$, which is on par with the state-of-the-art for online learners. Note that this result does not require Assumption 3.4 to hold, and we experimentally test the adaptive diversity tuning procedure in settings with noisy feedback in the next section.

## 5 EXPERIMENTAL STUDY

We consider several DPPs from the QDT family: QD decomposition, conditional DPP and the $\varepsilon$-greedy approach described in Section 3, along with baselines from the literature: Deep DPP (Gartrell et al., 2018), Markov DPP (Affandi et al., 2012), MMR (Carbonell & Goldstein, 1998), and xQuAD (Santos et al., 2010) (see Appendix D.3). To try to get as close as possible to a realistic online setting, we consider the following situation. At recommendation time $t > 0$, for a new user

---

[5] Analoguously to a feedback model, $\boldsymbol{y}$ is such that $\boldsymbol{y}(\phi, \boldsymbol{h})$ is the (observed) feedback from the user $\boldsymbol{h}$ if $\phi$ has been recommended, and otherwise is equal to 1 (meaning that it is ignored for a non-visited $\phi$).

$\boldsymbol{h}^t$ with ground-truth history $\boldsymbol{\mathcal{H}}^t$ from the initial data set, a recommender system will output a batch of recommendations for each round $t, t+1, \ldots, t+|\boldsymbol{\mathcal{H}}^t|$ with respective user context-history pairs $(\boldsymbol{h}^t, \emptyset), (\boldsymbol{h}^t, \{i_1\}), (\boldsymbol{h}^t, \{i_1, i_2\}), \ldots, (\boldsymbol{h}^t, \boldsymbol{\mathcal{H}}^t)$, where $\boldsymbol{\mathcal{H}}^t := \{i_1, i_2, \ldots, i_M\}$ if $|\mathcal{H}^t| = M$ in the initial data set. It means that at each round, we incrementally increase the user history with true previously recommended items, and track the relevance and diversity of novel recommendations (see all metrics in Section 1). As the SAMPLING strategy is not deterministic, we iterate this process over 10 random seeds, and average all the metrics across those 10 iterations for each user. For pointwise metrics such as in Equation 1, we also average these metrics along the trajectory of recommendations of length $|\boldsymbol{\mathcal{H}}^t|$. Since we run this setting on several users, we aggregate these metrics across users by considering the average and the standard deviation. These are the final values shown in the tables in this section. Top ones are in bold type, second best one are underlined. For the sake of clarity, we only show the most relevant metrics and baselines in experiments. Runtime is the time in seconds needed to output a single batch of recommendations. All values are rounded to the closest 2$^{\text{nd}}$ decimal place.

For all experiments, we consider a linear kernel function $k : \phi, \psi \mapsto \phi^\mathsf{T} \psi$ to compare fairly with baselines for the quality-diversity trade-off. Similarly, we use as default value $\lambda = 0.5$: equal weight for the quality and diversity tasks; $\varepsilon = 0.1$: frequency of the greedy phase, to get a strong baseline for diversity; and $\alpha = 0$: diversity hyperparameter for B-DivRec in Equation 5; unless otherwise specified. We consider the following data sets in our experiments: (1) Synthetic data sets: their names are prefixed with SYNTHETIC, followed by the number of items $N$. Supplementary information can be found in Appendix D.1; (2) MovieLens data set (Harper & Konstan, 2015) for movie recommendation, with feedback values in $1, 2, \ldots, 5$; Epinions data set (Leskovec et al., 2010) in social networks and a collection of six data sets for drug repurposing (Cdataset, DNdataset, Fdataset, Gottlieb, LRSSL, and PREDICT), with feedback values in $[0, 1]$. Further details about the real-life data sets are in Appendix D.2.

**Known feedback setting with fixed $\lambda$.** For the sake of readability, we postpone our sensitivity analyses on hyperparameters $\lambda$ and $\alpha$, and the comparison between the SAMPLING and MAXIMIZATION strategies to Appendix E. As expected, the larger $\lambda$ is, the more weight is put on the relevance task, and there is a global increase of the related metric when $\lambda$ is larger. Also unsurprisingly, a greater value of $\alpha$ leads to a higher global diversity div$^G$, but might impact relevance as well, highlighting the presence of the quality-diversity tradeoff. Moreover, we found the MAXIMIZATION strategy to better perform the expected quality-diversity tradeoff, and we will use this strategy for all DPP-based approaches from now on. We also test larger values of the batch size $B$ in Appendix F. However, note that we considered small batch sizes as ultimately in real-life applications, the user might only be willing to grade up to 5 items at a time. Moreover, we focus our study on real-life data sets and baselines out of the QDT family, pushing results related to synthetic data sets and the QDT family to Appendix E.

We now display a summary of the numerical results obtained on real-life data and on methods Deep DPP, MMR, xQuAD and B-DivRec in Tables 1-8. The full results on this benchmark can be found in Appendix D.3. On the MovieLens data set for movie recommendation, MMR is the top contender, whereas our contribution B-DivRec is the second best, improving significantly upon the conditional DPP in terms of relevance. The div$^G$ values are particularly small on MovieLens due to the fact that initial user histories are collinear: they feature similar movie embeddings, which leads to a small volume. The same phenomenon can be observed on the Epinions data set. We discuss this issue further in Appendix G. However, on all other data sets, B-DivRec achieves the quality-diversity tradeoff, with relevance values close to the quality-wise top performers MMR and xQuAD, while largely improving on the global diversity metric. This is confirmed by the fact that B-DivRec has the best value of effective diversity div$^+$ across all eight data sets but one.

**Adaptive quality-diversity trade-off.** Finally, we evaluate our adaptive procedure for tuning the quality-diversity trade-off parameter $\lambda \in [0, 1]$. Note that $\lambda$ is specific to one user: each online learner is initialized and updated along a trajectory corresponding to a single user. As such, we only consider one user, user with identifier 0, in all data sets. Then, we run B-DivRec (Equation 5) combined with the MAXIMIZATION strategy and the adaptive approach described in Section 4. We report the results in Table 9 for MovieLens and PREDICT. To assess the goodness of our approach, the final tuned value $\lambda_{\text{final}}$ is compared to the best *a posteriori* relevance weight $\lambda_\star$. This

Table 1: Benchmark on Cdataset (4 users, $B = 3$) with MAXIMIZATION.

| ALGO | REL ↑ | DIV$^G$ ↑ | DIV$^+$ ↑ |
|---|---|---|---|
| Deep DPP | 0.36 ±0.03 | 0.16 ±0.02 | 0.32 ±0.04 |
| MMR | **0.91 ±0.01** | 0.14 ±0.01 | 0.36 ±0.0 |
| xQuAD | **0.91 ±0.01** | 0.14 ±0.01 | 0.37 ±0.01 |
| **B-DivRec** | 0.79 ±0.01 | **0.22 ±0.02** | **0.52 ±0.02** |

Table 2: DNdataset (4 users, $B = 3$) with MAXIMIZATION.

| REL ↑ | DIV$^G$ ↑ | DIV$^+$ ↑ |
|---|---|---|
| 0.23 ±0.06 | 0.72 ±0.01 | 0.23 ±0.06 |
| **0.33 ±0.06** | 0.86 ±0.03 | 0.19 ±0.05 |
| **0.33 ±0.06** | 0.77 ±0.01 | 0.22 ±0.06 |
| 0.31 ±0.06 | **0.98 ±0.01** | **0.25 ±0.07** |

Table 3: Benchmark on Epinions (3 users, $B = 3$) with MAXIMIZATION. DeepDPP could not be run on Epinions, as the number of items is too large to compute the likelihood function in memory.

| ALGO | REL ↑ | DIV$^G$ ↑ | DIV$^+$ ↑ |
|---|---|---|---|
| Deep DPP | – | – | – |
| MMR | **0.04 ±0.01** | 0.0 ±0.0 | 0.0 ±0.0 |
| xQuAD | **0.04 ±0.01** | 0.0 ±0.0 | 0.0 ±0.0 |
| **B-DivRec** | 0.02 ±0.01 | 0.0 ±0.0 | 0.0 ±0.0 |

Table 4: Fdataset (4 users, $B = 3$) with MAXIMIZATION.

| REL ↑ | DIV$^G$ ↑ | DIV$^+$ ↑ |
|---|---|---|
| 0.48 ±0.03 | 0.12 ±0.01 | 0.59 ±0.06 |
| **0.95 ±0.01** | 0.16 ±0.01 | 0.35 ±0.01 |
| 0.93 ±0.01 | 0.14 ±0.01 | 0.37 ±0.01 |
| 0.81 ±0.01 | **0.24 ±0.02** | **0.64 ±0.05** |

Table 5: Benchmark on Gottlieb (4 users, $B = 3$) with MAXIMIZATION.

| ALGO | REL ↑ | DIV$^G$ ↑ | DIV$^+$ ↑ |
|---|---|---|---|
| Deep DPP | 0.50 ±0.03 | 0.19 ±0.02 | 0.55 ±0.05 |
| MMR | **0.94 ±0.01** | 0.37 ±0.03 | 0.64 ±0.02 |
| xQuAD | 0.93 ±0.01 | 0.36 ±0.03 | 0.65 ±0.01 |
| **B-DivRec** | 0.82 ±0.01 | **0.40 ±0.02** | **0.78 ±0.01** |

Table 6: LRSSL (4 users, $B = 3$) with MAXIMIZATION.

| REL ↑ | DIV$^G$ ↑ | DIV$^+$ ↑ |
|---|---|---|
| 0.39 ±0.02 | 0.53 ±0.05 | 0.64 ±0.03 |
| **0.97 ±0.0** | 0.55 ±0.05 | 0.82 ±0.02 |
| **0.97 ±0.0** | 0.53 ±0.04 | 0.80 ±0.02 |
| 0.83 ±0.01 | **0.56 ±0.04** | **0.86 ±0.01** |

Table 7: Benchmark on MovieLens (4 users, $B = 3$) with MAXIMIZATION. DeepDPP could not be run in full on MovieLens (1 iteration is shown), as it is very time-consuming: 500 seconds per iteration and user.

| ALGO | REL ↑ | DIV$^G$ ↑ | DIV$^+$ ↑ |
|---|---|---|---|
| Deep DPP | 1.24 ±0.07 | 0.02 ±0.01 | 0.22 ±0.05 |
| MMR | 3.78 ±0.12 | **0.07 ±0.01** | **0.77 ±0.04** |
| xQuAD | **3.85 ±0.15** | 0.05 ±0.01 | 0.68 ±0.04 |
| **B-DivRec** | 3.48 ±0.13 | 0.06 ±0.01 | 0.59 ±0.05 |

Table 8: PREDICT (4 users, $B = 3$) with MAXIMIZATION.

| REL ↑ | DIV$^G$ ↑ | DIV$^+$ ↑ |
|---|---|---|
| 0.21 ±0.02 | 0.35 ±0.03 | 0.30 ±0.04 |
| **0.79 ±0.02** | 0.49 ±0.03 | 0.54 ±0.03 |
| **0.79 ±0.02** | 0.47 ±0.03 | 0.64 ±0.02 |
| 0.68 ±0.03 | **0.52 ±0.03** | **0.80 ±0.03** |

Table 9: Benchmark with and without ("non adaptive") the adaptive tuning procedure in Section 4, applied to B-DivRec (Equation 5) with the MAXIMIZATION strategy for user 0, starting with initial value $\lambda_0 = 0.5$. $\lambda_{\text{final}}$: final relevance weight after tuning. $\lambda_\star$: best *a posteriori* relevance weight. We also test the adaptive procedure with noisy/observed feedback ("noisy", violating Assumption 3.4), without access to the feedback model $q_\Theta$. Further details on the noise for each data set can be found in Appendix C.3.

| DATA SET | $\lambda_{\text{final}}$ | $\lambda_\star$ | REL ↑ | DIV$^G$ ↑ | DIV$^+$ ↑ | TIME ↓ |
|---|---|---|---|---|---|---|
| **SYNTHETIC750** | 0.18 | 0.10 | 0.61 ±0.0 | 0.95 ±0.0 | 1.01 ±0.0 | 0.5 ±0.0 |
| non adaptive | – | – | 0.61 ±0.0 | 0.96 ±0.0 | 1.01 ±0.0 | 0.16 ±0.0 |
| *noisy, adaptive* | 0.73 | 0.5 | 1.00 ±0.0 | 0.16 ±0.0 | 0.07 ±0.02 | 1.50 ±0.11 |
| *noisy, non adaptive* | – | – | 1.00 ±0.0 | 0.16 ±0.0 | 0.07 ±0.02 | 0.06 ±0.0 |
| **MovieLens** | 1.00 | 1.00 | 4.49 ±0.01 | 0.01 ±0.0 | 0.77 ±0.01 | 158.1 ±0.84 |
| non adaptive | – | – | 4.38 ±0.0 | 0.01 ±0.0 | 0.78 ±0.0 | 29.7 ±0.82 |
| *noisy, adaptive* | 1.00 | 1.00 | 5.00 ±0.0 | 0.02 ±0.0 | 0.94 ±0.01 | 57.56 ±15.55 |
| *noisy, non adaptive* | – | – | 2.57 ±0.26 | 0.02 ±0.0 | 0.94 ±0.0 | 2.72 ±0.01 |
| **PREDICT** | 0.35 | 0.10 | 0.79 ±0.01 | 0.49 ±0.0 | 0.72 ±0.01 | 1.97 ±0.05 |
| non adaptive | – | – | 0.79 ±0.01 | 0.49 ±0.0 | 0.72 ±0.01 | 0.82 ±0.0 |
| *noisy, adaptive* | 0.97 | 0.99 | 0.41 ±0.04 | 0.21 ±0.03 | 0.54 ±0.02 | 7.7 ±0.17 |
| *noisy, non adaptive* | – | – | 0.28 ±0.05 | 0.33 ±0.03 | 0.83 ±0.01 | 1.36 ±0.08 |

value is computed at the end of a trajectory associated with one user by solving the maximization problem in $\lambda$ in Equation 6, and corresponds to the best deterministic action that the player could have taken in the game with Nature explained in Section 4. First, the adaptive procedure seems to be able to roughly retrieve the oracle value $\lambda_\star$ across data sets–albeit it is unlikely that it can always find it, since the oracle value relies on the *a posteriori* knowledge of the user feedback. Second, the computational cost of using the adaptive procedure instead of a fixed value of $\lambda = 0.5$ is moderate: the runtime with the adaptive procedure is multiplied by 8 in average across all data sets, due to the approximation of matrix power (see Appendix B). Third, on both MovieLens and PREDICT, adaptively selecting $\lambda$ throughout the trajectory allows us to noticeably increase relevance while trading off some diversity. MovieLens, since the users seem to be more biased towards popular movies compared to novel recommendations–as illustrated by the low value of div$^G$ in the non-adaptive noiseless setting–the recommender system is leaning toward making popular recommendations, leading to a higher $\lambda$, higher relevance (rel $+2.5\%$) to the price of some of effective diversity (div$^+$ $-1.3\%$).

## 6 DISCUSSION

In this paper, we introduced a versatile and flexible approach for diverse recommendation. We proposed a general family of DPPs, named the QDT family, embedding the quality-diversity trade-off which includes several well-known recommender systems. Then we introduced B-DivRec to integrate user history in a computationally tractable fashion. We illustrated the versatility of B-DivRec when the feedback model is known, and when the recommender system must adapt the diversity of its recommendations to the user based on user interaction with the recommended items.

Yet, many venues for research are still unresolved in the field. Recommender systems often face a pervasive issue related to missing feedback. How can one leverage information from the fact that a user has not visited nor rated a recommended item? This issue is connected with overcoming Assumption 3.4 on noiseless feedback (Radlinski et al., 2008; Zenati et al., 2022; Hikmawati et al., 2024; Park & Jia, 2025). However, improving upon the QDT framework might be the key to solve the issue of missing feedback while still ensuring the diversity of recommended items. A first approach could implement the optimism principle as in multi-armed bandits (Auer et al., 2002; Abbasi-Yadkori et al., 2011) and building confidence intervals on the expected feedback on items in the likelihood matrix of a DPP from the QDT family. Another venue of research is to adapt the rotting bandit framework (Levine et al., 2017) to incorporate global diversity.

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

## A RELATED WORKS (CONTINUED)

Due to space constraints in the main text, we write here a more comprehensive section than in Section 2 about prior works on recommender systems with diversity.

### A.1 SETTING OF THIS PAPER AND B-DIVREC

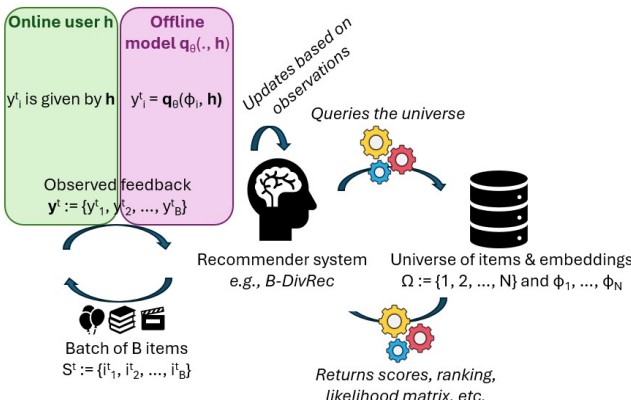

Figure 1: Setting of our paper.

The objective of our paper is to study and optimize the tradeoff between accuracy and different levels of diversity (local and global) in recommendations of batches of items. B-DivRec, as all other considered baselines, is applied directly on the (whole) library of items and outputs batches of recommended items, hence the importance of tractable computations. Moreover, its formulation exploits the estimated feedback values from a backbone (feedback) model. The strength of our approach is that we can consider any backbone model $q_\Theta$ for the feedback, as long as it satisfies Assumptions 3.1-3.3 as formulated in the paper. Note that those assumptions are unrestrictive in practice.

Moreover, as we do not have access to an online setting where we could directly interrogate users, we had to set up an offline framework on observed ratings for real-life data sets: *e.g.*, SVD for MovieLens, heterogeneous graph attention network for PREDICT, trained offline on observed data, as described in Appendix D.2.

### A.2 DIVERSITY METRICS

In Section 1, we focused on a geometrically intuitive definition of diversity relying on the volume of the parallelotope built by the columns of the kernel function applied to a given subset of items. That definition is versatile enough that, not only could we use it to define intrabatch/local and interbatch/global diversity metrics (Bederina & Vie, 2025) (see Equation 1), but it also enabled us to produce a single metric for the quality-diversity trade-off (that is, the div$^+$ metric in Equation 2). These metrics hold for any feedback model and kernel function, hence they are very flexible. As described in Section 2, prior works (Carbonell & Goldstein, 1998; Kaminskas & Bridge, 2016) also proposed a linear or convex combination of relevance and diversity metrics, controlled by a parameter $\lambda \leq 0$, or a quality-diversity trade-off induced by a matrix decomposition (Kulesza & Taskar, 2010). In our paper, we actually combine the two approaches. We discuss below other metrics mentioned in the literature. Note that we make a distinction between metrics which are only evaluated in the experimental studies (*e.g.*, intralist average distance or category coverage (Ge et al., 2010; Kaminskas & Bridge, 2016; Chen et al., 2018; Li et al., 2024)) from the metrics which are optimized upon and actually used in the recommender systems. We focus on the latter in this section.

A strong competitor of our metrics are ridge leverage scores. Given the ridge hyperparameter $\zeta \geq 0$, for a given kernel function $k$, Musco & Musco (2017, Definition 1) define the ridge leverage score of item $\phi^i \in \Omega := \{\phi^1, \phi^2, \ldots, \phi^N\}$ as $r_k^\zeta(\phi^i) := (L(L + \zeta I_N)^{-1})_{i,i}$, where $L := k_{\Omega,\Omega}$. Alternatively, defining $U \in \mathbb{R}^{N \times d}$ such that $K = UU^\intercal$, the ridge leverage score for $\phi^i$ is the

value of the ridge regression problem $\min_{\boldsymbol{y} \in \mathbb{R}^N} \|U_{i,\cdot} - \boldsymbol{y}U\|_2^2 + \zeta\|\boldsymbol{y}\|_2^2 \in [0, 1]$. In other words, $\boldsymbol{y}$ is the mixing vector that achieves the linear combination of rows in $U$ closest to the $i^{\text{th}}$ row of $U$ (which is the row associated with $\boldsymbol{\phi}^i$). Then the ridge leverage score is a measure of the unicity of the vector $\boldsymbol{\phi}^i$ among elements in $\Omega$: the smaller it is, the least "unique" $\boldsymbol{\phi}^i$ is. However, computing the exact ridge leverage scores requires potentially inverting a matrix of size $N \times N$ which itself has a time complexity of $\mathcal{O}(N^3)$. To get a measure of diversity for a whole set of items $\mathcal{S}$, summing all leverage scores across $\mathcal{S}$ yields the "effective dimension", also called "degrees of freedom": $d_{\text{eff},k}^{\zeta}(\mathcal{S}) := \sum_{\boldsymbol{\phi}^i \in \mathcal{S}} r_k^{\zeta}(\boldsymbol{\phi}^i) = \text{tr}(L(L + \zeta I_N)^{-1})$. This measure of diversity is quite intuitive, and the time complexity is in $\widetilde{\mathcal{O}}(N)$, where $\widetilde{\mathcal{O}}$ hides logarithmic terms in $N$ (Chen & Yang, 2021). Then, similarly to determinants and volumes where Cholesky decompositions on sparse or low-rank matrices can be leveraged (see Appendix B), the time complexity of these measures might be linear in the number of items. However, since the determinant is directly optimized (in the MAXIMIZATION strategy) or considered (in the SAMPLING strategy) in our algorithms, an evaluation of recommender systems by the volume might be more consistent than by leverage scores.

As mentioned in Section 1, the topic of the quality-diversity trade-off is related to serendipity–which is also called out-of-the-box (Abbassi et al., 2009), unexpectedness (Xu & Matsumura, 2024), surprise (Ziarani & Ravanmehr, 2021)–where an unexpectedly good item is recommended to a user, contrary to simply recommending items which are very different from those in the user history or cover different categories (which would be diversity). Looking for serendipitous items allows us to overcome the popularity bias in recommendation (Yu et al., 2024). Kaminskas & Bridge (2016); Poulain & Tarissan (2020); Ziarani & Ravanmehr (2021); Kotkov et al. (2024) show that there is no consensus regarding the concept of serendipity, nor a single metric. The same holds for diversity metrics: recently, Mironov & Prokhorenkova (2024) tried to formalize the expected properties of a good diversity metric, showed that known diversity metrics do not match them, and proposed two matching metrics. These properties are monotonicity (the diversity of the union of two sets should be higher than the maximum diversity for each of these sets), uniqueness (replacing an element from a set with a copy of an already present element should decrease the diversity of the modified set), and continuity of the diversity metric. In particular, the determinant does not satisfy the monotonicity property. Yet the metrics that Mironov & Prokhorenkova (2024) proposed are too computationally expensive in practice. In Yu et al. (2024), authors built a recommender system for long-tail Web services/items likely to be queried by applications/users, which is based on linear propagation of information on graphs of interconnected services with a LightGCN (He et al., 2020). This allows them to learn item and user embeddings. The predicted probability of querying a service by an application is the inner product of the corresponding embeddings. A recommender system is then trained on those predicted scores with a pairwise Bayesian Personalized Ranking (BPR) loss, which is appropriate with implicit and sparse feedback data. Other papers also study diversity through random walks on graphs (Poulain & Tarissan, 2020). Contrary to our paper, most of the reported metrics combine metrics of item unpopularity (related to relevance) and item dissimilarity (Iaquinta et al., 2008; Abbassi et al., 2009) to be applied at a reranking stage where a subset of relevant items has already been retrieved. However, as reported in Li et al. (2024), filtering too early for relevant items might cut down diversity in an irreversible way. Moreover, albeit the presence of numerous novelty-related metrics, the selection of a single metric is subjective, whereas the quality and diversity metrics are usually straightforwardly guided by the recommendation task.

### A.3 ABOUT DISTRIBUTIONS ON RANDOM SUBSETS OF POINTS

This section focuses on theoretical developments in DPPs which are relevant to our paper.

We recall that selecting $B$ elements among $N$ using a determinantal point process (DPP) has a time complexity of $\mathcal{O}(NB^3)$, after finding the eigenvalues of the likelihood matrix, which has a time complexity of $\mathcal{O}(N^3)$. In the case of a linear kernel on $d$-dimensional item embeddings, finding the eigenvalues has a time complexity of $\mathcal{O}(Nd^2)$. Moreover, using the dual representation of DPPs, sampling $B$ elements among $N$ has a time complexity of $\mathcal{O}(NdB^2 + d^2B^3)$ (Kulesza et al., 2012), whereas the greedy approximation used for the maximization strategy (which is a NP-hard problem otherwise) has a naive time complexity of $\mathcal{O}(B^2N)$. However, several works (Gillenwater et al., 2012; Han et al., 2017; Chen et al., 2018) improve upon this time complexity, by leveraging the fact that the logarithm of the determinant of the likelihood matrix in the DPP should be maximized

(see Equation 4) and using linear algebra approximations. $\log \det$ is a submodular function but not monotone; instead of obtaining a $1 - 1/e$ approximation of the exact solution (Kempe et al., 2003), proposed approaches yield an $1/4$ approximation (Gillenwater et al., 2012), and run with a time complexity in $\mathcal{O}(NB^2)$ (Chen et al., 2018). Moreover, Mariet et al. (2019) introduce an approximation of a DPP–without the quality-diversity trade-off–by a deep learning model, named DPPNet, which could also be used to further empirically speed up the sampling and maximization of a DPP. Implementing our contributions with a DPPNet instead of a DPP might be a future venue for enhancing their use in real-life applications.

A recent paper by Kawashima & Hino (2024) introduces a family of distributions on sets of points which includes DPPs, called discrete kernel point processes (DKPPs), where the attraction and the repulsion of sampled points can be controlled explicitly. Given a continuous function $\psi : \mathbb{R}^+ \mapsto \mathbb{R}$ and $k_{\Omega,\Omega}$ a positive-semidefinite Hermitian matrix, the probability of sampling the subset $A \subseteq \Omega$ is given by $\mathbb{P}(\mathcal{S} = A) \propto \exp\left(\mathrm{tr}(\psi(k_{A,A}))\right)$. $\psi$ controls the parametrization between positive and negative correlations. DKPPs enable maximization, *i.e.*, finding the subset with highest probability, and sampling as well. However, using DPPs allows us to leverage existing implementations and algorithms. Furthermore, positive correlations between items are not necessarily linked to relevance, as the user information or context might also intervene in the quality score.

## A.4 BASELINES FOR DIVERSIFIED RECOMMENDATIONS

This section gives an overview of the state-of-the-art on diversified recommender systems. We focused on Determinantal Point Processes (DPPs) in our paper to leverage the literature regarding fast implementation of sampling and maximization algorithms, and to incorporate some flexibility in the description of item similarity thanks to the use of kernel functions. Here, we do not consider prior works where the feedback model needs to be learned on the fly (Radlinski et al., 2008; Chao et al., 2015; Kathuria et al., 2016; Nava et al., 2022; Hikmawati et al., 2024; Park & Jia, 2025).

As mentioned in Section 1, DPPs are popular in the field of recommender systems. Recently, Ibrahim et al. (2025) applied two DPPs (a traditional quality-diversity decomposition, and one with a linear kernel, without quality scores nor trade-off parameter) to output relevant and diverse recommendations of cultural goods to teenagers, again, in the sense of "intrabatch" diversity. They reported an improved diversity (volume) in the recommendation with DPPs, to the price of a notable part of the relevance, both in offline and online/live experiments. Wilhelm et al. (2018) (using a quality-diversity decomposition with a RBF kernel) also applied DPPs on video recommendation. Authors showed that DPPs, contrary to all other baselines, yielded an increase in the number of long user sessions, which are indicative of user satisfaction with the recommendations.

Recommender systems that build upon MMR, which is a strong baseline in the field of diverse recommendations, have also been investigated in the literature. Recently, Li et al. (2024) introduced Contextual Distillation Model (CDM), which trains a surrogate attention-based deep model of Maximal Marginal Relevance (MMR), for a diversity metric correlated to the scalar product of item and user vectors. The surrogate model estimates the probability of a given item of being ranked among the top-$B$ items by the MMR score (see Section 2). Authors compared their contribution to a DPP (Chen et al., 2018) and MMR, showing that the DPP had a better performance diversity-wise, but arguing that the latter have a quadratic time complexity in the number of items $N$. However, we showed in our work that, provided some approximations (see Appendix B), we can actually run DPPs (B-DivRec and the conditional DPP) and MMR on libraries of dozens of millions of items. Note that Li et al. (2024) only focused on what we call "intrabatch diversity".

In addition to (Affandi et al., 2012), there are other papers regarding the improvement of the diversity compared to a past sequence of recommended items. In addition to proposing a faster algorithm for the MAXIMIZATION strategy for DPPs, Chen et al. (2018) also studies the case when the diversity is only required among items in a sliding time window. They use a standard quality-diversity decomposition for the likelihood matrix, with the MAXIMIZATION strategy, and modify the maximization problem such that only the last $\omega - 1$ recommended items are taken into account in the greedy algorithm. However, their approach is suitable when a long series of interactions with the user occurs and when we allow the recommender system to forget the oldest recommendations. Yet, in practice, users interact only few times with the recommender system before settling on a recommendation or dropping out (Ben-Porat et al., 2022; Gusak et al., 2025).

Finally, few papers on the literature deal with the issue of tuning the level of diversity in the recommendations to the user. Given a notion of usefulness (*i.e.*, relevance) and unexpectedness, which is somewhat related to diversity (see the previous paragraph). Xu & Matsumura (2024) studies the problem of finding the good proportions of usefulness and unexpectedness in recommendations for each user after a step of retrieval of relevant items. The latter is closely related to the challenge of addressing the adaptive quality–diversity trade-off in Section 4, albeit our procedure operates at any stage of recommendation (including retrieval) and does not feature a serendipity metric. The metric in Xu & Matsumura (2024) incorporates for each user a convex combination of long-term and short-term preferences as the usefulness-unexpectedness trade-off ("curiosity") parameter. In that paper, this parameter is precomputed for each user, and cannot change anymore, unlike our approach.

## B    PRACTICAL IMPLEMENTATION

We resorted to the Python package DPPy (Gautier et al., 2019; Calandriello et al., 2020) for the implementation of the SAMPLING strategy, whereas we used the greedy algorithm in (Chen et al., 2018) for the MAXIMIZATION approach. Please refer to the `requirements.txt` file in the code for the package versions. This section is motivated by the fact that, in Algorithm 1, the step of computation of the likelihood matrix for the DPP in Line 5 and of the recommendation set in Lines 6-7 might be expensive (at least in $\mathcal{O}(N^3)$) when $N$ is large when implemented naively.

We also used sparse matrices as implemented in the Python package SciPy (Virtanen et al., 2020) with a denuding procedure: meaning that we could make the matrix sparser by rounding up values in matrices up to the $q^{\text{th}}$ decimal place to save memory. In practice, for our experiments, we did not have to round up values. However, for even larger libraries of items, this technique might prove useful. Below, we list the most computationally expensive parts of the code and explain the solutions considered to get a tractable implementation.

### B.1    NYSTRÖM APPROXIMATIONS OF THE KERNEL FUNCTION

One important element of the code is that a dense matrix of size $N \times N$ should never be stored in full in the RAM. To comply with this rule, we extensively relied on the feature map $\nu$ associated with a kernel function $k$ (which also facilitates the computations as if $k$ were a linear function). To compute $\nu$, we use a Nyström approximation (Nyström, 1930) that assumes that the rank of $k_{\Omega,\Omega}$ is actually $C \ll N$. Given a dimension $d' \ll d$, the Nyström approximation starts by subsampling a set of $C$ items $\mathcal{I}$ in $\Omega$ at random (without replacement), and then builds the eigendecomposition of the real-valued matrix $k_{\Omega,\Omega}$, that is, the evaluation of the kernel function on the whole universe.

$$k_{\Omega,\Omega} \quad = \quad U\Lambda U^\intercal = \begin{bmatrix} U_\mathcal{I}\Lambda U_\mathcal{I}^\intercal & U_\mathcal{I}\Lambda U_{\neg\mathcal{I}}^\intercal \\ U_{\neg\mathcal{I}}\Lambda U_\mathcal{I}^\intercal & U_{\neg\mathcal{I}}\Lambda U_{\neg\mathcal{I}}^\intercal \end{bmatrix} = \begin{bmatrix} k_{\mathcal{I},\mathcal{I}} & k_{\mathcal{I},\neg\mathcal{I}} \\ k_{\mathcal{I},\neg\mathcal{I}}^\intercal & k_{\neg\mathcal{I},\neg\mathcal{I}} \end{bmatrix},$$

where $U_\mathcal{I}$ (respectively, $U_{\neg\mathcal{I}}$) is the part of the orthonormal basis of the eigendecomposition associated with items in $\mathcal{I}$ (resp., not in $\mathcal{I}$), and $\Lambda$ is the diagonal matrix of eigenvalues of $k_{\Omega,\Omega}$. When one needs to compute $k_{\mathcal{S},\mathcal{S}}$ for any subset $\mathcal{S} \subset \Omega$ of size $B$, using the fact that $k_{\mathcal{S},\mathcal{S}} = \left(k_{\mathcal{S},\mathcal{I}}^\intercal U_\mathcal{I}\Lambda^{-1/2}\right)\left(k_{\mathcal{S},\mathcal{I}}^\intercal U_\mathcal{I}\Lambda^{-1/2}\right)^\intercal$ by replacing $U_{\neg\mathcal{I}}$ by $k_{\neg\mathcal{I},\mathcal{I}}U_\mathcal{I}\Lambda^{-1}$. $\Lambda^{-1/2}$ consists in simply applying the square root function to the eigenvalues of $k_{\Omega,\Omega}$, whereas $k_{\mathcal{S},\mathcal{I}} \in \mathbb{R}^{B \times d'}$ where $d' \ll N$ can be explicitly evaluated. All in all, the corresponding $\nu$ function is $\nu(\mathcal{S}) = k_{\mathcal{S},\mathcal{I}}^\intercal U_\mathcal{I}\Lambda^{-1/2}$ for any $\mathcal{S} \subset \Omega$.

As previously mentioned in the main text, the time complexity of the Nyström approximation is $\mathcal{O}(N(d')^2 + (d')^3)$. In practice, we use the implementation of the Nyström approximation in the Python package scikit-learn (Pedregosa et al., 2011). The computation of $\mathcal{I}$, $U_\mathcal{I}$ and $\Lambda$ happens only once before any recommendation is made. We use $d' = 100$ for the rank in the Nyström approximation. We note that, while we use random elements to build the approximation, other papers suggest to consider more representative points (Tremblay et al., 2019), for instance, those with maximum leverage score which would be more representative of the set. However, this approach can be costly (see Appendix A) which is why we elected to stick to random sampling.

Note that the computation of the volume (see metrics in Equation 1) also relies on the Nyström approximation, as the full matrix $k_{\mathcal{S},\mathcal{S}}$ is not built. This accounts for the fact that, although Assumption 3.3 mentions that the item embeddings have a $\ell_2$-norm equal to 1, the actual computed volumes in the experiments in Section 5 might exceed 1. The Nyström approximation does not preserve the property on the norm of the item embeddings.

### B.2    APPROXIMATE MATRIX INVERSIONS AND DETERMINANTS

For the conditional and the k-Markov (Affandi et al., 2012) DPPs, there is a step of inversion of a positive definite matrix of potentially size $N \times N$, which is extremely costly in a naive implementation. As suggested by prior works (Burian et al., 2003), we resort to Cholesky decompositions of the matrices that we wish to inverse. The Cholesky decomposition of a positive-definite matrix $M$ of size $n$ is $M = R^\intercal R$ where $R$ is an upper (or lower) triangular matrix of size $n$. Then, as

$M^{-1} = R^{-1}(R^{-1})^\intercal$, it suffices to solve $RX = I_n$ in $X$ by forward substitution, because $R$ is a triangular matrix, to obtain the inverse of $M$. To compute the (log-)determinant of $M$ of size $n$, the following relation is true: $\log \det(M) = 2 \sum_{i=1}^{n} \log R_{i,i}$, because $R$ is a triangular matrix with positive diagonal elements (Madar, 2015) and $\det R = \det R^\intercal$.

In our implementation, we use the Python package Scikit-Sparse. For the conditional DPP, the Nyström approximation of the kernel function can be leveraged on top of the Cholesky decomposition (decomposing $\widetilde{A}^{1/2}\widetilde{A}^{1/2} \approx A$ instead of $A$, which is less expensive due to the low-rank assumption for $\widetilde{A}^{1/2}$), which is why the conditional DPP can be run on large data sets contrary to Markov DPP. Indeed, the theoretical time complexity of the Cholesky decomposition is in $\mathcal{O}(N^3)$ as a general rule. However, for sparse matrices, a fill-reducing Cholesky decomposition can be computed by reordering rows to restrict the creation of new nonzero elements (Duff & Uçar, 2009), and for sparse matrices, the practical time complexity is typically much smaller than cubic in $N$.

### B.3 APPROXIMATE MATRIX POWER

Matrix power intervenes in Equation 3 which describes the likelihood matrix for a DPP of the QDT family for $\lambda \neq 1/2$, where the power is possibly real-valued. The exact computation would require computing the whole set of eigenvalues and eigenvectors, which is quickly intractable as $N$ increases. Similarly to Appendix B.1, we make a low-rank assumption and consider the truncated Singular Value Decomposition (SVD) of rank $r \ll N$ of a real-valued matrix $M$ where $M$ is a square real matrix of size $N$. The SVD yields $M = U\Lambda V^\intercal$, where $U, V \in \mathbb{R}^{r \times N}$ are orthogonal matrices. Then we compute $V\Lambda^p V^\intercal$ as a proxy for $M^p$, that is, $M$ multiplied $p$ times where $p \in \mathbb{R}^+$. Indeed, $M^p \approx (V^\intercal \Lambda V)^p = \underbrace{(V^\intercal \Lambda V)(V^\intercal \Lambda V) \dots (V^\intercal \Lambda V)}_{p \text{ times}} = V^\intercal \Lambda^p V$, since $VV^\intercal = I_r$

and $V^\intercal V = I_N$. This operation has a time complexity in $\mathcal{O}(Nr^2)$ instead of the naive time complexity $\mathcal{O}(N^3)$ which is dominated by the computation of the full eigendecomposition. Moreover, if we need to find $L$ instead, where $LL^\intercal = M^p$, we can output $L = V\sqrt{\Lambda^p}$ where $\sqrt{\Lambda^p}$ is such that $\sqrt{\Lambda^p}\sqrt{\Lambda^p} = \Lambda^p$. Note that, for the diagonal matrix $\Lambda^p$, $\sqrt{\Lambda^p}$ is the result of the square root function applied element-wise to the diagonal elements of $\Lambda^p$. We choose $r = N - 1$ if $N \leq 1\,000$ and $r = 100$ otherwise.

### B.4 APPROXIMATE CLOSEST-NEIGHBOR FINDING ALGORITHMS

B-DivRec requires the computation of the maximum cosine distance between feature maps of items from the current batch and the user history (see Equation 5). Obviously, computing all cosine distances between any item and all items in the user history would be expensive since it is performed for all items in $\Omega$. As mentioned in Section 3, many approaches allow us to retrieve the closest neighbor of a point in a set. Our implementation uses FAISS Douze et al. (2024), which is suitable for large data sets. In practice, we do not even need to build a new tree for each user and each recommendation round, but only a single tree on all items in $\Omega$. Indeed, queries to FAISS can be made while ignoring a subset of items when querying for the closest neighbor (in our case, ignoring all items which are not in the current user's history).

### B.5 DIVERSITY ISSUES IN DPPS

As mentioned in Section 5, beyond a certain value of $\alpha$, B-DivRec can no longer sample enough diverse items, as $\alpha$ filters out most of the items. This issue is not specific to our contribution, as Ibrahim et al. (2025) has also reported it for other conditional DPPs. It arises when the rank of the likelihood matrix ($L$-matrix) of the DPP is strictly smaller than $B$.

Possible solutions to bypass this issue would be (1) to return a batch of recommended items of size smaller than $B$ (with a risk of no longer recommending any item at some point), (2) to only focus on quality and set the matrix $f(k, \mathcal{S}, \mathcal{H}) \leftarrow I_N$, (3) to forget some of the user history, and only preserve either the most representative points or the latest recommendations. Future work on this topic would be of interest for practical recommender systems.

## B.6 RETRIEVING EMBEDDINGS

Finally, our implementation cannot afford to store the full item embedding matrix $\Phi = [\phi^i, i \in \Omega] \in \mathbb{R}^{N \times d}$ in the RAM. First, we store the user histories in files, and update them in-place. Second, we store, access and read the item embedding matrix by batches in memory with .csv files. Moreover, for the largest data set of our paper (the synthetic data set SYNTHETIC15M with five millions items), we use the .npy binary format instead of .csv for faster input/output operations.

# C  ADAPTIVE QUALITY-DIVERSITY TRADE-OFF

We recall here the online learning game for the adaptive quality-diversity trade-off (see Section 4). The player is the recommender system, and Nature is the interaction with *one* user $(\boldsymbol{h}, \mathcal{H}^t)$ querying the recommender system at time $t$ (Auer et al., 2002; De Rooij et al., 2014). At round $t \leq T$ of the game, the recommender system chooses a value of $\lambda^t \in [0, 1]$. Then, the recommender system makes a recommendation $\mathcal{S}^t$ to user $\boldsymbol{h}$ and receives a loss value related to the quality-diversity trade-off achieved, depending on the vector of feedback $\boldsymbol{y}^t$ from Nature. The recommender system should use this loss to update and use $\lambda^{t+1} \in [0, 1]$ at the next round. The game ends after $T$ interactions with the same user.

## C.1  IMPLEMENTATION OF THE ADAPTIVE DIVERSITY-TUNING PROCEDURE

Applying the AdaHedge algorithm (De Rooij et al., 2014) to our problem, we consider two experts. One favors diversity, whereas the other prefers quality. We denote the posterior weight vector associated with those two experts $\boldsymbol{\lambda}^t := [1 - \lambda^t, \lambda^t]$ at time $t$. In our case, we consider a gain instead of a loss function–since the optimization problem in Equation 6 is a maximization problem. The function (to maximize) at time $t$ with $\lambda^t$ is

$$
\begin{aligned}
\mathrm{scr}_{f,\boldsymbol{y}^t}^{\lambda^t}(\mathcal{S}^t; \boldsymbol{h}^t) &= 4(1 - \lambda^t) \log \mathrm{vol}(f(k, \mathcal{S}^t, \mathcal{H}^t)) + 4\lambda^t \log \det\left(\mathrm{diag}(\{y_k^t\}_{k \leq B})\right) \quad (7) \\
&= 4(1 - \lambda^t) \log \mathrm{vol}(f(k, \mathcal{S}^t, \mathcal{H}^t)) + 4\lambda^t \sum_{k \leq B} \log y_k^t,
\end{aligned}
$$

where $f = f^{\text{B-DivRec}}$ (note however that this approach straightforwardly works for any DPP of the QDT family as described in Equation 3). Now, since $\mathrm{scr}_{f,\boldsymbol{y}^t}^{\lambda^t}(\mathcal{S}^t; \boldsymbol{h}^t)$ is clearly linear in $\lambda^t$ (hence concave), we can apply the gradient trick as follows

$$
\begin{aligned}
\mathcal{R}^{\text{adapt}}(T; \boldsymbol{h}) &:= \max_{\lambda \in [0,1]} \sum_{t \leq T} \mathrm{scr}_{f,\boldsymbol{y}^t}^{\lambda}(\mathcal{S}^t; \boldsymbol{h}) - \mathrm{scr}_{f,\boldsymbol{y}^t}^{\lambda^t}(\mathcal{S}^t; \boldsymbol{h}) \\
&\leq \max_{\lambda \in [0,1]} \sum_{t \leq T} (\lambda - \lambda^t) \nabla_\lambda \mathrm{scr}_{f,\boldsymbol{y}^t}^{\lambda^t}(\mathcal{S}^t; \boldsymbol{h}).
\end{aligned}
$$

The gradient $\nabla_\lambda \mathrm{scr}_{f,\boldsymbol{y}^t}^{\lambda}(\mathcal{S}^t; \boldsymbol{h})$ with respect to $\lambda \in [0, 1]$ is straightforward to compute

$$
\nabla_\lambda \mathrm{scr}_{f,\boldsymbol{y}^t}^{\lambda}(\mathcal{S}^t; \boldsymbol{h}^t) := -4 \log \mathrm{vol}(f(k, \mathcal{S}^t, \mathcal{H}^t)) + 4 \sum_{k \leq B} \log y_k^t. \quad (8)
$$

Then, we obtained an upper bound on the quantity that we want to optimize in the form of a inner product. This is the motivation for using as a gain function at time $t$

$$
g_t : \boldsymbol{\lambda} \in \triangle_2 \mapsto [-\nabla_\lambda \mathrm{scr}_{f,\boldsymbol{y}^t}^{\lambda}(\mathcal{S}^t; \boldsymbol{h}^t), \nabla_\lambda \mathrm{scr}_{f,\boldsymbol{y}^t}^{\lambda}(\mathcal{S}^t; \boldsymbol{h}^t)],
$$

where $\triangle_2$ is the simplex of dimension 2. Note that the first coordinate of $g_t$ $g_t(\boldsymbol{\lambda})_1$ is actually the gradient of $\mathrm{scr}_{f,\boldsymbol{y}^t}^{\lambda}(\mathcal{S}^t; \boldsymbol{h})$ with respect to $1 - \lambda$ (proven with a change of variable). Then in the AdaHedge learner, at time $t$, we update the posterior weight with

$$
-\langle \boldsymbol{\lambda}^t, g_t(\boldsymbol{\lambda}) \rangle = -\left((1 - \lambda^t)(-g_t(\boldsymbol{\lambda})_2) + \lambda^t g_t(\boldsymbol{\lambda})_2\right) = (1 - 2\lambda^t) g_t(\boldsymbol{\lambda})_2 = (1 - 2\lambda^t) \nabla_\lambda \mathrm{scr}_{f,\boldsymbol{y}^t}^{\lambda}(\mathcal{S}^t; \boldsymbol{h}^t),
$$

which corresponds to Line 8 in Algorithm 1. Then, we obtain a new value $\boldsymbol{\lambda}^{t+1} \in \triangle_2$, from which we extract the second coordinate $\lambda^{t+1}$ for the next round.

## C.2  UPPER BOUND ON THE REGRET

**Theorem C.1.** Upper bound on the regret incurred by the adaptive diversity tuning procedure (Theorem 4.1 in the main text). *An upper bound on the regret $\mathcal{R}^{adapt}(T; \boldsymbol{h})$ (Equation 6) incurred by the adaptive strategy for tuning the level of diversity $\lambda \in [0, 1]$ for user $\boldsymbol{h}$ over $T$ rounds of recommendations is*

$$
\mathcal{R}^{adapt}(T; \boldsymbol{h}) \leq 2\delta_T \sqrt{T \log(2)} + 16\delta_T(2 + \log(2)/3),
$$

*where $\delta_T := 8 \max_{t \leq T} \log \frac{M_t^B}{a_t}$, $a_t := vol(f(k, S^t, H^t))$ and $M_t := \max_{i \leq B} y_i^t$.*

*Proof.* We defined the regret $\mathcal{R}^{\mathrm{adapt}}(T; \boldsymbol{h})$ for user $\boldsymbol{h}$ at time $T$ in Equation equation 6, with the gain function $G_t : \lambda \in [0,1] \mapsto \mathrm{scr}^{\lambda}_{f^{\text{B-DivRec}}, \boldsymbol{y}^t}(\mathcal{S}^t; \boldsymbol{h})$ for any $t \leq T$. $G_t$ is defined in Equation 4 and is linear (and then concave) in $\lambda$ for all $t \leq T$. Moreover, remember that we applied AdaHedge with the gain function $g_t$ at time $t$ in Line 8 in Algorithm 1, where

$$\forall \lambda \in [0,1], \quad g_t(\lambda) := (1 - 2\lambda) \underbrace{\left( -4\log(\mathrm{vol}(f(k, S^t, H^t))) + 4\sum_{i \leq B} \log y_i^t \right)}_{=C_t}.$$

by using the computation from Subsection C.1. We denote $\boldsymbol{\lambda} := [1 - \lambda, \lambda]^{\intercal} \in \triangle_2$, where $\triangle_2$ is defined as the simplex of dimension 2 $\{\boldsymbol{p} \in [0,1]^2 \mid p_1 + p_2 = 1\}$. Then we can rewrite the expression of $g_t$ as

$$\forall \lambda \in [0,1], \quad g_t(\lambda) = -\langle \boldsymbol{\lambda}, [C_t, -C_t]^{\intercal} \rangle.$$

Then, combining the concavity of $G_t$ in $\lambda$, the fact that $C_t$ is constant in $\lambda$, and the definition of $g_t$,

$$\forall \lambda \in [0,1], \sum_{t \leq T} G_t(\lambda) - G_t(\lambda^t) \quad \leq \quad \sum_{t \leq T} \langle \boldsymbol{\lambda} - \boldsymbol{\lambda}^t, [C_t, -C_t]^{\intercal} \rangle$$

$$\implies \mathcal{R}^{\mathrm{adapt}}(T; \boldsymbol{h}) \quad \leq \quad \max_{\lambda \in [0,1]} \sum_{t \leq T} g_t(\lambda) - g_t(\lambda^t).$$

Then, we apply (De Rooij et al., 2014, Theorem 8 and Corollary 17) to the loss function $-g_t$, using the fact that $\lambda \in [0,1]$,

$$R^{\mathrm{adapt}}(T; \mathbf{h}) \leq 2\delta_T \sqrt{T \log(2)} + 16\delta_T(2 + \log(2)/3), \quad \text{where } \delta_T := 2\max_{t \leq T} \max(C_t, -C_t).$$

What remains is to evaluate $\delta_T$. According to Assumptions 3.1-3.3, there are two positive constants $m_t$ and $M_t \geq 1$ such that $0 < m_t \leq y_i^t \leq M_t$, and $0 < a_t := \mathrm{vol}(f(k, S^t, H^t)) \leq 1$. Using the computations in Subsection C.1, the following inequalities hold

$$\forall t \leq T, \quad -4\log a_t + 4B\log m_t \quad \leq \quad C_t \leq -4\log a_t + 4B\log M_t$$
$$4\log(m_t^B/a_t) \quad \leq \quad C_t \leq 4\log(M_t^B/a_t).$$

This means that $\delta_T \leq 8\max_{t \leq T} \log(\max(m_t, M_t)^B/a_t) = 8\max_{t \leq T} \log \frac{M_t^B}{a_t}$. $\qquad \square$

Note that Assumption 3.2 can be replaced by an assumption on the positiveness of *observed* feedback values, hence this result still hold true when we do not have access to a feedback model.

## C.3 EXPERIMENTS WITH NOISY FEEDBACK

We discuss in Section 5 in the main text our testing approach. This section focuses on testing the adaptive procedure under noisy feedback, hence violating Assumption 3.4. We also add results obtained on SYNTHETIC30k (under noiseless feedback). We replace estimated feedback values by model $q_\Theta$ with observed–positive, due to Assumption 3.2–feedback.

We first add noise in the synthetic data set. The recommender system no longer receives the exact feedback score $q_\Theta(\boldsymbol{\phi}^i, \boldsymbol{h}^t)$ at time $t$, but a binary outcome in $\{1, 2\}$ determined by a Bernouilli law of probability $p := \min(1, \max(0, q_\Theta(\boldsymbol{\phi}^i, \boldsymbol{h}^t)))$. Note that 1 is drawn with probability $1 - p$, and 2 with probability $p$. This setting simulates the case where we only receive clicks, and no longer probabilities of clicks. We report below the results compared to B-DivRec applied to the same data set without adaptive tuning of $\lambda$, starting from $\lambda^0 = 0.5$. As the relevance is already close to the maximum (which is 2) in this case, there is no observable improvement when running the adaptive procedure.

We also consider the setting where the recommender system only has access to the outcomes of clinical trials (3: successful, 2: not tested, 1: unsuccessful) from the PREDICT data set, or to

Table 10: Benchmark with and without ("non adaptive") the adaptive tuning procedure in Section 4, applied to B-DivRec (Equation 5) with the MAXIMIZATION strategy for user 0, starting with initial value $\lambda_0 = 0.5$. $\lambda_{\text{final}}$: final relevance weight after tuning. $\lambda_\star$: best *a posteriori* relevance weight. We also test the adaptive procedure with noisy feedback ("noisy", violating Assumption 3.4). We renormalize the average feedback values to match the ones from the original feedback model: removing 1 to feedback values in SYNTHETIC750 and MovieLens, substracting 2 to feedback values in PREDICT.

| DATA SET | $\lambda_{\text{final}}$ | $\lambda_\star$ | REL ↑ | DIV$^{\text{G}}$ ↑ | DIV$^+$ ↑ | TIME ↓ |
|---|---|---|---|---|---|---|
| SYNTHETIC750 | 0.18 | 0.10 | 0.61 ±0.0 | 0.95 ±0.0 | 1.01 ±0.0 | 0.5 ±0.0 |
| (non adaptive) | – | – | 0.61 ±0.0 | 0.96 ±0.0 | 1.01 ±0.0 | 0.16 ±0.0 |
| (noisy, adaptive) | 0.73 | 0.5 | 1.00 ±0.0 | 0.16 ±0.0 | 0.07 ±0.02 | 1.50 ±0.11 |
| (noisy, non adaptive) | – | – | 1.00 ±0.0 | 0.16 ±0.0 | 0.07 ±0.02 | 0.06 ±0.0 |
| SYNTHETIC30k | 0.19 | 0.10 | 0.65 ±0.0 | 0.96 ±0.0 | 1.0 ±0.0 | 29.8 ±0.06 |
| (non adaptive) | – | – | 0.56 ±0.0 | 0.96 ±0.0 | 1.01 ±0.0 | 1.34 ±0.02 |
| MovieLens | 1.0 | 1.0 | 4.49 ±0.01 | 0.01 ±0.0 | 0.77 ±0.01 | 158.1 ±0.84 |
| (non adaptive) | – | – | 4.38 ±0.0 | 0.01 ±0.0 | 0.78 ±0.0 | 29.7 ±0.82 |
| (noisy, adaptive) | 0.997 | 0.997 | 5.00 ±0.0 | 0.02 ±0.0 | 0.94 ±0.01 | 57.56 ±15.55 |
| (noisy, non adaptive) | – | – | 2.57 ±0.26 | 0.02 ±0.0 | 0.94 ±0.0 | 2.72 ±0.01 |
| PREDICT | 0.35 | 0.10 | 0.79 ±0.01 | 0.49 ±0.0 | 0.72 ±0.01 | 1.97 ±0.05 |
| (non adaptive) | – | – | 0.79 ±0.01 | 0.49 ±0.0 | 0.72 ±0.01 | 0.82 ±0.0 |
| (noisy, adaptive) | 0.974 | 0.993 | 0.41 ±0.04 | 0.21 ±0.03 | 0.54 ±0.02 | 7.7 ±0.17 |
| (noisy, non adaptive) | – | – | 0.28 ±0.05 | 0.33 ±0.03 | 0.83 ±0.01 | 1.36 ±0.08 |

the movie ratings from 1 (not seen), 2 (rated 1 star) to 6 (five stars) from the MovieLens data set. In both PREDICT and MovieLens, similarly to what we described in the paper, we observe an improvement of the relevance (to the price of some of the diversity values) when using the adaptive tuning procedure, and the procedure is able to almost retrieve the best a posteriori value $\lambda^\star$ from the starting value $\lambda^0 = 0.5$. In particular, in MovieLens, the algorithm almost always returns top-rated movies in the successive 233 batches of 3 recommended items for user 0.

As a general rule, considering noisy feedback expectedly decreases the relevance values, but this effect can be mitigated when using the adaptive procedure, which is another argument in favor of this approach in real-life settings. These experiments validate our findings even when we only have access to observable feedback (and not to the estimated feedback values by a proxy model). The adaptive procedure allows us to improve on the relevance, when possible. When the relevance cannot be improved further, the relevance-wise and diversity-wise performance remains the same as in the non-adaptive setting.

## D SUPPLEMENTARY MATERIAL ON THE EXPERIMENTAL STUDY

All experiments have been run on a remote server (configuration: processor Intel Xeon Processor (Skylake, IBRS), 12 cores @2.4GHz, RAM 23GB).

### D.1 GENERATION OF SYNTHETIC DATA SETS

The synthetic data sets aim at providing an easy approach to test the scalability of the algorithms and the diversity in recommendations. The key idea is that $d$-dimensional item embeddings $(\phi^1, \ldots, \phi^N)$ and user contexts $(h^1, \ldots, h^{N_u})$ are sampled at random (hence $m = d$), and the feedback value for the item-user pair $(\phi, h)$ is $((\phi)^\intercal h + 1)/2 \in [0, 1]$. We actually generate $B$ groups of items, with the expectation that, when outputting a batch of $B$ recommendations, a good recommender system for diversity would sample elements from each group.

Given a desired number of items $N$ and a batch size $B$, we first sample at random and independently $\lceil N/B \rceil \times d$ values from $\mathcal{N}(0, 2)$ to build a first matrix of $\ell_2$-normalized item embeddings $\Phi^1 := [\phi^1, \phi^2, \ldots, \phi^{\lceil N/B \rceil}] \in \mathbb{R}^{\lceil N/B \rceil \times d}$. Then, for the $\ell^{\text{th}}$ group, $\ell = 2, \ldots, B$, we construct the matrix $\Phi^\ell := \Phi^i + 0.01\ell \mathbf{1}_{\lceil N/B \rceil \times d}$, where $\mathbf{1}_{n \times d}$ is the matrix of size $n \times d$ which coefficients are ones. Finally, we concatenate all (renormalized) matrices $\Phi^1, \Phi^2, \ldots, \Phi^B$ to obtain $\Phi \in \mathbb{R}^{N \times d}$. For users, given a maximum number of users $N_u$, $N_u \times d$ values are sampled from $\mathcal{N}(0, 1)$ to build the user contexts, after $\ell_2$-normalization.

### D.2 TRAINING ON REAL-LIFE DATA SETS

In this section, we delve more into detail concerning the two real-life data sets for movie recommendation and drug repurposing. In both cases, the important part is to learn the feedback model $q_\Theta$ based on prior user-item interactions (*e.g.*, previously rated movies or terminated clinical trials, collected offline by browsing clinical trial registries). The feedback model is then used in the computation of the likelihood matrix in Line 5 in Algorithm 1. See the `README.md` file for instructions to download the files related to those data sets.

**Movie recommendation: MovieLens.** The MovieLens data set (Harper & Konstan, 2015) that we considered has 9,725 items, 610 users and 100,837 nonzero ratings in $\{1, 2, \ldots, 5\}$. The higher the rating, the more the user liked the movie. Metadata about the items/movies are the movie title, year of release, and the genre tags (*e.g.*, comedy, action). First, 512-dimensional item embeddings of norm 1 are built by applying Universal Sentence Encoder (Cer et al., 2018) to the movie metadata, using Python package Tensorflow Hub (Abadi et al., 2016). Second, we split the list of item-user pairs with nonzero ratings into a training set (75%) and a testing set (25%) at random. We fit a SVD model on the training set using the Python package Mangaki (Vie et al., 2015), which fills the zeroes in the ratings matrix of MovieLens. We computed the Root Mean Squared Error on the testing set, yielding RMSE= 0.96. Knowing that the ratings are in $\{1, 2, \ldots, 5\}$, it means that, in average, the mistake made by the SVD model on the predicted rating is not enough to accidentally classify a very bad movie as a good one. We considered a classification threshold at $\tau = 2.5$.

**Trust network: Epinions.** The Epinions data set (Leskovec et al., 2010) contains users' ratings (from 1 to 5) on items. We use the same approach as for MovieLens to define the feedback model. To make it run smoothly across all baselines, we only kept users with at least 50 ratings, and items with at least 100 ratings, and used as item embedding matrix the first factor of the SVD, for a total of 160,417 items.

**Drug repurposing: , Cdataset (Luo et al., 2016), DNdataset (Martinez et al., 2015), Fdataset (Gottlieb et al., 2011), Gottlieb (Gao et al., 2022), LRSSL (Liang et al., 2017), and PREDICT (Réda, 2023).** Those data sets comprise information about drugs and diseases, along with the status of Phase 2 or 3 clinical trials involving each drug-disease pair: 0 means that the pair has not been tested in a Phase 2 or 3 clinical trial, 1 means that the clinical trial was successful in showing that the drug has a therapeutic effect on the disease, and -1 means that the clinical trial failed (*e.g.*, low accrual, emergence of adverse side effects). There are six types of drug features and five types of disease features across all data sets, corresponding to the similarity of a drug (respectively,

Table 11: Datasets in the benchmark. They correspond to the number of drugs and diseases involved in at least one nonzero drug-disease association. The rating matrix in the `Fdataset` comes from Gottlieb et al. (2011), whereas the drug and disease features are from Luo et al. (2016).

| NAME. | N | $d$ | #USERS $m$ | | #ONES | #MINUS ONES |
|-------|-----|-------|------|-------|-------|-------------|
| Cdataset | 663 | 663 | 409 | 409 | 2,532 | 0 |
| DNdataset | 550 | 1,490 | 360 | 4,516 | 1,008 | 0 |
| Fdataset | 593 | 593 | 313 | 313 | 1,933 | 0 |
| Gottlieb | 593 | 1,779 | 313 | 313 | 1,933 | 0 |
| LRSSL | 763 | 2,049 | 681 | 681 | 3,051 | 0 |
| PREDICT | 1,014 | 1,642 | 941 | 1,490 | 4,627 | 132 |

a disease) to another drug (resp., disease) in the data set with respect to each type of feature. We give the description and relevant metrics regarding each drug repurposing data set in Table 11. We fitted the Heterogeneous Attention Network (HAN) algorithm (Wang et al., 2019) as feedback model to those data sets, using the Python package stanscofi Réda et al. (2024), splitting nonzero ratings at random into training ($80\%$) and testing ($20\%$) sets. The Area Under the Curve on the testing set for the PREDICT data set (the richest data set in terms of features) was $0.92$.

### D.3 BENCHMARK ON XQUAD AND DEEP DPP

xQuAD (Santos et al., 2010) relies on the definition of subqueries in the ranking score for items. This ranking score is similar to the one for MMR (Carbonell & Goldstein, 1998), except for not usng any diversity metrics, but considering relevance scores for items fitting each subquery. This means in particular that xQuAD only explicitly relies on the feedback model $q_\Theta$. To make the comparison fairer with other baselines relying on the diversity metrics, we consider as sub-query generation procedure the selection of the items in the history with a cosine similarity higher than $\alpha$ (similarly to what was defined for our algorithm B-DivRec). To implement this, we resort to FAISS trees as described in Equation 5. Then, the main difference between xQuAD and BDivRec is that xQuAD only considers the relevance scores of items and similar items in the history in its ranking score.

Deep DPP (Gartrell et al., 2018) learns a low-rank factor $V$ of the likelihood matrix for a DPP, meaning that the final likelihood matrix is $L = VV^\intercal$. This learning is based on sets of observed subsets of items, and backpropagation of the likelihood function. In our benchmark, we define the observed subsets needed for learning the likelihood matrix as batches of items of at most $B$ in the user history. Contrary to B-DivRec, Deep DPP does not include the estimated feedback values nor the diversity values in the ranking score. Moreover, if the user history is empty, we set the likelihood matrix to the identity matrix.

We perform tests on a synthetic data set with 1,500 items (SYNTHETIC1500), the Epinions data set (Leskovec et al., 2010), and on all drug repurposing data sets in Appendix D.2. Those results are respectively displayed in Tables 12-19. The results are somewhat expected, as xQuAD do not take into account explicitly the similarity scores between items and the corresponding algorithm has almost the same structure as MMR. Deep DPP leverages the power of DPPs to perform well (local) diversity-wise, but is worst at relevance. BDivRec clearly improves upon all baselines either in terms of diversity (xQuAD) or relevance (Deep DPP), highlighting the quality-diversity tradeoff we aimed for. However, note that xQuAD and Deep DPP were not developed for the optimization of the quality-global diversity tradeoff.

Table 12: Benchmark on SYNTHETIC1500 (4 users, $B = 5$). REL: relevance. PREC: precision with $\tau = 0.5$. $\text{DIV}^L$: intrabatch/local diversity. $\text{DIV}^G$: interbatch/global diversity. $\text{DIV}^+$: effective diversity. TIME: runtime.

| MAXIMIZ. | REL ↑ | PREC ↑ | $\text{DIV}^L$ ↑ | $\text{DIV}^G$ ↑ | $\text{DIV}^+$ ↑ | TIME ↓ |
|---|---|---|---|---|---|---|
| Deep DPP | 0.50 ±0.0 | 0.47 ±0.01 | **1.00** ±0.0 | 0.32 ±0.0 | 0.98 ±0.0 | 0.70 ±0.01 |
| **B-DivRec** | 0.53 ±0.0 | 0.82 ±0.02 | **1.00** ±0.0 | **0.94** ±0.00 | **1.01** ±0.00 | 0.06 ±0.00 |
| **NO DPP** | | | | | | |
| MMR | **0.65** ±0.0 | **1.00** ±0.0 | 0.00 ±0.0 | 0.00 ±0.0 | 0.00 ±0.0 | **0.03** ±0.00 |
| xQuAD | **0.65** ±0.0 | **1.00** ±0.0 | 0.00 ±0.0 | 0.00 ±0.0 | 0.00 ±0.0 | 0.76 ±0.01 |

Table 13: Benchmark on Epinions (3 users, $B = 3$). REL: relevance. PREC: precision with $\tau = 0.5$. $\text{DIV}^L$: intrabatch/local diversity. $\text{DIV}^G$: interbatch/global diversity. $\text{DIV}^+$: effective diversity. TIME: runtime.

| MAXIMIZ. | REL ↑ | PREC ↑ | $\text{DIV}^L$ ↑ | $\text{DIV}^G$ ↑ | $\text{DIV}^+$ ↑ | TIME ↓ |
|---|---|---|---|---|---|---|
| **B-DivRec** | 0.02 ±0.01 | 0.0 ±0.0 | 0.0 ±0.0 | 0.0 ±0.0 | 0.0 ±0.0 | 17.15 ±0.56 |
| **NO DPP** | | | | | | |
| MMR | **0.04** ±0.01 | 0.0 ±0.0 | 0.0 ±0.0 | 0.0 ±0.0 | 0.0 ±0.0 | **3.01** ±0.15 |
| xQuAD | **0.04** ±0.01 | 0.0 ±0.0 | 0.0 ±0.0 | 0.0 ±0.0 | 0.0 ±0.0 | 51.94 ±3.11 |

# E   COMPLEMENTARY EXPERIMENTS

We report and describe here the sensitivity analyses on hyperparameters $\lambda$ and $\alpha$, and the comparison between the SAMPLING and MAXIMIZATION strategies.

First, we perform a sensitivity analysis on hyperparameters $\lambda \in [0, 1]$ across the QDT family and baselines with the MAXIMIZATION strategy on the SYNTHETIC750 data set, and $\alpha \in [0, 2]$ for B-DivRec (Equation 5). The trends for relevance and diversity depending on $\lambda$ and $\alpha$ are shown respectively in Figure 2 and Table 21. As expected, across all recommender systems, as $\lambda$ increases, relevance increases, whereas (local or global) diversity globally decreases. The higher the area under the curve (AUC), the better the recommender system. Visually, the top two recommender systems are our contribution B-DivRec and MMR. Computing explicitly the areas under the curve for each metric (rounding to the closest third decimal place) shows that B-DivRec outperforms MMR on all metrics except for the global diversity. Overall, members of the QDT family outperform non-DPP baselines such as MMR and Markov DPP, especially for precision and relevance. Regarding $\alpha$ in B-DivRec, unsurprisingly, the higher $\alpha$, the more (globally) diverse the recommendations, to the price of a loss in relevance and precision. Beyond a certain value of $\alpha$ (for this data set, $\alpha = 1$), the DPP can no longer sample enough diverse items, as $\alpha$ filters out most of the items. We discuss this issue and potential solutions in Appendix B.

Next, we apply both the SAMPLING and MAXIMIZATION strategies for all recommender systems on the SYNTHETIC750 data set in Tables 22-23. The values of relevance for SAMPLING and of precision for MAXIMIZATION are not reported because they are similar across all algorithms but MMR (respectively, rel $= 0.51$, and prec $= 1.0$ for all algorithms except for Markov DPP, where prec $= 0.94$, see the full results in Table 20). These results confirm that, for the quality-diversity trade-off, B-DivRec with the MAXIMIZATION strategy and MMR are the top contenders. Moreover, they show that the SAMPLING indeed encourages (global) diversity as evidenced by the $\text{div}^G$ values and as reported in prior work (Kathuria et al., 2016). However, it results in a large loss in precision (prec) and relevance (rel), it is more time-consuming, and does not quite achieve the best quality-diversity trade-off ($\text{div}^+$). The SAMPLING strategy might be useful in the

Table 14: Benchmark on Cdataset (4 users, $B = 3$). REL: relevance. PREC: precision with $\tau = 0.5$. $\text{DIV}^{\text{L}}$: intrabatch/local diversity. $\text{DIV}^{\text{G}}$: interbatch/global diversity. $\text{DIV}^{+}$: effective diversity. TIME: runtime.

| MAXIMIZ. | REL ↑ | PREC ↑ | $\text{DIV}^{\text{L}}$ ↑ | $\text{DIV}^{\text{G}}$ ↑ | $\text{DIV}^{+}$ ↑ | TIME ↓ |
|---|---|---|---|---|---|---|
| Deep DPP | 0.36 ±0.03 | 0.35 ±0.05 | 0.48 ±0.01 | 0.16 ±0.02 | 0.32 ±0.04 | 0.37 ±0.02 |
| **B-DivRec** | 0.79 ±0.01 | 0.98 ±0.0 | **0.50 ±0.02** | **0.22 ±0.02** | **0.52 ±0.02** | **0.25 ±0.0** |
| **NO DPP** | | | | | | |
| MMR | **0.91 ±0.01** | **1.00 ±0.0** | 0.36 ±0.0 | 0.14 ±0.01 | 0.36 ±0.0 | 0.35 ±0.01 |
| xQuAD | **0.91 ±0.01** | **1.00 ±0.0** | 0.37 ±0.01 | 0.14 ±0.01 | 0.37 ±0.01 | 0.54 ±0.02 |

Table 15: Benchmark on DNdataset (4 users, $B = 3$). REL: relevance. PREC: precision with $\tau = 0.5$. $\text{DIV}^{\text{L}}$: intrabatch/local diversity. $\text{DIV}^{\text{G}}$: interbatch/global diversity. $\text{DIV}^{+}$: effective diversity. TIME: runtime.

| MAXIMIZ. | REL ↑ | PREC ↑ | $\text{DIV}^{\text{L}}$ ↑ | $\text{DIV}^{\text{G}}$ ↑ | $\text{DIV}^{+}$ ↑ | TIME ↓ |
|---|---|---|---|---|---|---|
| Deep DPP | 0.23 ±0.06 | 0.24 ±0.06 | 0.81 ±0.01 | 0.72 ±0.01 | 0.23 ±0.06 | 1.58 ±0.08 |
| **B-DivRec** | 0.31 ±0.06 | **0.25 ±0.07** | **1.01 ±0.0** | **0.98 ±0.01** | **0.25 ±0.07** | **1.37 ±0.04** |
| **NO DPP** | | | | | | |
| MMR | **0.33 ±0.06** | **0.25 ±0.07** | 0.89 ±0.03 | 0.86 ±0.03 | 0.19 ±0.05 | 1.57 ±0.07 |
| xQuAD | **0.33 ±0.06** | **0.25 ±0.07** | 0.82 ±0.01 | 0.77 ±0.01 | 0.22 ±0.06 | 1.61 ±0.07 |

case when the recommender system must not recommend the same batch for users with the same embeddings and history, and when we are willing to recommend possibly irrelevant items. However, when the focus is on the optimization of the quality-diversity trade-off and computational efficiency, the MAXIMIZATION strategy might be more suitable.

We also ran the same type of experiments on SYNTHETIC3M and SYNTHETIC15M in Tables 24 and 25. Beyond 5 000 items, MarkovDPP (Affandi et al., 2012) is actually computationally intractable to run, hence we left it out of the benchmark for larger data sets. The observations made on SYNTHETIC750 (superiority of the MAXIMIZATION strategy over the SAMPLING one in terms of relevance) from those experiments are confirmed on larger synthetic data sets. Moreover, even if MMR might have better results on relevance-related metrics (rel, prec), this baseline clearly fails for diversity metrics ($\text{div}^{\text{G}}$, $\text{div}^{+}$), especially on larger data sets. As the number of items $N$ increases, the difference in performance between algorithms from the QDT family decreases. This might be due to the fact that the more items there are, the less important is the impact of user history on the selection of items. We also applied recommender systems to MovieLens in Table 7, and PREDICT in Table 8. We only show in these tables the results with the MAXIMIZATION strategy, on account of the observations on the synthetic data sets.

Table 16: Benchmark on Fdataset (4 users, $B = 3$). REL: relevance. PREC: precision with $\tau = 0.5$. DIV$^L$: intrabatch/local diversity. DIV$^G$: interbatch/global diversity. DIV$^+$: effective diversity. TIME: runtime.

| MAXIMIZ. | REL ↑ | PREC ↑ | DIV$^L$ ↑ | DIV$^G$ ↑ | DIV$^+$ ↑ | TIME ↓ |
|---|---|---|---|---|---|---|
| Deep DPP | 0.48 ±0.03 | 0.55 ±0.05 | 0.47 ±0.01 | 0.12 ±0.01 | 0.59 ±0.06 | 0.28 ±0.01 |
| **B-DivRec** | 0.81 ±0.01 | **1.0 ±0.0** | **0.64 ±0.05** | **0.24 ±0.02** | **0.64 ±0.05** | **0.19 ±0.01** |
| **NO DPP** | | | | | | |
| MMR | **0.95 ±0.01** | **1.0 ±0.0** | 0.35 ±0.01 | 0.16 ±0.01 | 0.35 ±0.01 | 0.27 ±0.01 |
| xQuAD | 0.93 ±0.01 | **1.0 ±0.0** | 0.37 ±0.01 | 0.14 ±0.01 | 0.37 ±0.01 | 0.44 ±0.01 |

Table 17: Benchmark on Gottlieb (4 users, $B = 3$). REL: relevance. PREC: precision with $\tau = 0.5$. DIV$^L$: intrabatch/local diversity. DIV$^G$: interbatch/global diversity. DIV$^+$: effective diversity. TIME: runtime.

| MAXIMIZ. | REL ↑ | PREC ↑ | DIV$^L$ ↑ | DIV$^G$ ↑ | DIV$^+$ ↑ | TIME ↓ |
|---|---|---|---|---|---|---|
| Deep DPP | 0.50 ±0.03 | 0.58 ±0.06 | 0.64 ±0.01 | 0.19 ±0.02 | 0.55 ±0.05 | **0.44 ±0.01** |
| **B-DivRec** | 0.82 ±0.01 | 0.97 ±0.02 | **0.76 ±0.01** | **0.40 ±0.02** | **0.78 ±0.01** | **0.44 ±0.01** |
| **NO DPP** | | | | | | |
| MMR | **0.94 ±0.01** | **1.00 ±0.0** | 0.64 ±0.02 | 0.37 ±0.03 | 0.64 ±0.02 | 0.72 ±0.02 |
| xQuAD | 0.93 ±0.01 | 0.99 ±0.0 | 0.65 ±0.01 | 0.36 ±0.03 | 0.65 ±0.01 | 0.88 ±0.02 |

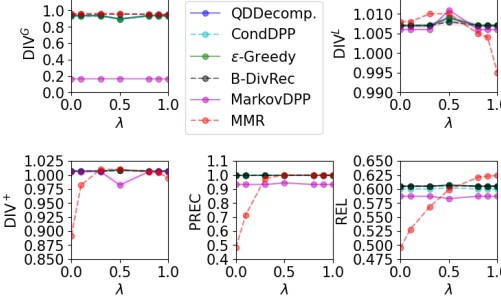

Figure 2: Sensitivity analysis for $\lambda$ on SYNTHETIC750 (6 users, $B = 3$, $\tau = 0.5$), with the MAXIMIZATION strategy.

Table 21: Sensitivity analysis for $\alpha$ on SYNTHETIC750 (6 users, $B = 3$, $\tau = 0.5$), with the MAXIMIZATION strategy.

| $\alpha$ | REL ↑ | PREC ↑ | DIV$^G$ ↑ |
|---|---|---|---|
| 0 | 0.61 ±0.0 | 1.0 ±0.0 | 0.95 ±0.0 |
| 0.1 | 0.61 ±0.0 | 1.0 ±0.0 | 0.95 ±0.0 |
| 0.5 | 0.61 ±0.0 | 1.0 ±0.0 | 0.95 ±0.0 |
| 0.8 | 0.61 ±0.0 | 1.0 ±0.0 | 0.96 ±0.0 |
| 0.90 | 0.6 ±0.0 | 1.0 ±0.0 | 0.97 ±0.0 |
| 0.95 | 0.58 ±0.0 | 1.0 ±0.0 | 0.97 ±0.0 |
| 0.98 | 0.57 ±0.0 | 0.97 ±0.0 | 0.97 ±0.0 |
| 1 | 0.56 ±0.0 | 0.92 ±0.01 | 0.96 ±0.0 |

Table 22: Benchmark on SYNTHETIC750 (6 users, $B = 3$, $\tau = 0.5$) with SAMPLING.

Table 23: SYNTHETIC750 (6 users, $B = 3$, $\tau = 0.5$) with MAXIMIZATION.

| ALGO | PREC ↑ | DIV$^G$ ↑ | DIV$^+$ ↑ | REL ↑ | DIV$^G$ ↑ | DIV$^+$ ↑ |
|---|---|---|---|---|---|---|
| QDDec. | 0.61 ±0.01 | 0.95 ±0.0 | **1.01 ±0.0** | **0.61 ±0.0** | 0.9 ±0.01 | **1.01 ±0.0** |
| CondDPP | 0.64 ±0.01 | 0.94 ±0.01 | 0.97 ±0.01 | 0.6 ±0.0 | 0.89 ±0.01 | **1.01 ±0.0** |
| $\varepsilon$-Greedy | 0.61 ±0.01 | 0.95 ±0.0 | **1.01 ±0.0** | **0.61 ±0.0** | 0.9 ±0.01 | **1.01 ±0.0** |
| Markov | 0.59 ±0.01 | 0.17 ±0.0 | 0.91 ±0.01 | 0.58 ±0.0 | 0.17 ±0.0 | 0.98 ±0.01 |
| MMR | **1.0 ±0.0** | **0.96 ±0.0** | **1.01 ±0.0** | 0.6 ±0.0 | **0.96 ±0.0** | **1.01 ±0.0** |
| **B-DivRec** | 0.63 ±0.01 | 0.95 ±0.0 | 0.98 ±0.01 | **0.61 ±0.0** | 0.95 ±0.0 | **1.01 ±0.0** |

Table 18: Benchmark on LRSSL (4 users, $B = 3$). REL: relevance. PREC: precision with $\tau = 0.5$. DIV$^L$: intrabatch/local diversity. DIV$^G$: interbatch/global diversity. DIV$^+$: effective diversity. TIME: runtime.

| MAXIMIZ. | REL ↑ | PREC ↑ | DIV$^L$ ↑ | DIV$^G$ ↑ | DIV$^+$ ↑ | TIME ↓ |
|---|---|---|---|---|---|---|
| Deep DPP | 0.39 ±0.02 | 0.40 ±0.03 | **0.90 ±0.01** | 0.53 ±0.05 | 0.64 ±0.03 | 0.48 ±0.02 |
| **B-DivRec** | 0.83 ±0.01 | **1.00 ±0.0** | 0.86 ±0.01 | **0.56 ±0.04** | **0.86 ±0.01** | **0.36 ±0.01** |
| **NO DPP** | | | | | | |
| MMR | **0.97 ±0.0** | **1.00 ±0.0** | 0.82 ±0.02 | 0.55 ±0.05 | 0.82 ±0.02 | 0.53 ±0.02 |
| xQuAD | **0.97 ±0.0** | **1.00 ±0.0** | 0.80 ±0.02 | 0.53 ±0.04 | 0.80 ±0.02 | 0.71 ±0.02 |

Table 19: Benchmark on PREDICT (4 users, $B = 3$). REL: relevance. PREC: precision with $\tau = 0.5$. DIV$^L$: intrabatch/local diversity. DIV$^G$: interbatch/global diversity. DIV$^+$: effective diversity. TIME: runtime.

| MAXIMIZ. | REL ↑ | PREC ↑ | DIV$^L$ ↑ | DIV$^G$ ↑ | DIV$^+$ ↑ | TIME ↓ |
|---|---|---|---|---|---|---|
| Deep DPP | 0.21 ±0.02 | 0.17 ±0.03 | 0.64 ±0.03 | 0.35 ±0.03 | 0.30 ±0.04 | 0.73 ±0.03 |
| **B-DivRec** | 0.68 ±0.03 | 0.78 ±0.04 | **0.86 ±0.03** | **0.52 ±0.03** | **0.80 ±0.03** | **0.50 ±0.02** |
| **NO DPP** | | | | | | |
| MMR | **0.79 ±0.02** | **0.92 ±0.02** | 0.56 ±0.03 | 0.49 ±0.03 | 0.54 ±0.03 | 0.76 ±0.04 |
| xQuAD | **0.79 ±0.02** | **0.92 ±0.02** | 0.66 ±0.03 | 0.47 ±0.03 | 0.64 ±0.02 | 1.00 ±0.06 |

## F  VARYING BATCH SIZE AND COMPUTATIONAL RUNTIME

We also performed supplementary experiments to test our setting. Note that we considered small batch sizes (*e.g.*, $B = 3$, $B = 5$ for SYNTHETIC15M) as ultimately in real-life applications, the user might only be willing to grade up to 5 items at a time. However, we run an experiment with B-DivRecand MMR on SYNTHETIC1500, with varying batch sizes. We report the average execution time in seconds over 10 iterations in Table 26. Albeit MMR is slightly faster for smaller values of batches, it becomes ×12 slower as the batch size increases.

Table 24: Benchmark on SYNTHETIC3M (4 users, $B = 3$) with MAXIMIZATION.

Table 25: SYNTHETIC15M (2 users, $B = 5$) with MAXIMIZATION.

| ALGO | REL ↑ | DIV$^G$ ↑ | DIV$^+$ ↑ | REL ↑ | DIV$^G$ ↑ | DIV$^+$ ↑ |
|---|---|---|---|---|---|---|
| CondDPP | 0.55 ±0.0 | **0.97 ±0.0** | **1.01 ±0.0** | 0.53 ±0.0 | **0.94 ±0.0** | **1.01 ±0.0** |
| MMR | **0.73 ±0.0** | 0.0 ±0.0 | 0.02 ±0.0 | **0.73 ±0.0** | 0.0 ±0.0 | 0.0 ±0.0 |
| **B-DivRec** | 0.55 ±0.0 | **0.97 ±0.0** | **1.01 ±0.0** | 0.53 ±0.0 | **0.94 ±0.0** | **1.01 ±0.0** |

Table 20: Benchmark on SYNTHETIC750 (6 users, $B = 3$). REL: relevance. PREC: precision with $\tau = 0.5$. DIV$^L$: intrabatch/local diversity. DIV$^G$: interbatch/global diversity. DIV$^+$: effective diversity. TIME: runtime.

| SAMPLING | REL ↑ | PREC ↑ | DIV$^L$ ↑ | DIV$^G$ ↑ | DIV$^+$ ↑ | TIME ↓ |
|---|---|---|---|---|---|---|
| QDDecomp. | 0.51 ±0.0 | 0.61 ±0.01 | **1.01 ±0.0** | 0.95 ±0.0 | **1.01 ±0.0** | 0.06 ±0.0 |
| CondDPP | 0.51 ±0.0 | 0.64 ±0.01 | 1.0 ±0.0 | 0.94 ±0.01 | 0.97 ±0.01 | 0.07 ±0.0 |
| $\varepsilon$-Greedy | 0.51 ±0.0 | 0.61 ±0.01 | 1.01 ±0.0 | 0.95 ±0.0 | 1.01 ±0.0 | 0.06 ±0.0 |
| MarkovDPP | 0.51 ±0.0 | 0.59 ±0.01 | 1.0 ±0.0 | 0.17 ±0.0 | 0.91 ±0.01 | 0.08 ±0.0 |
| **B-DivRec** | 0.51 ±0.0 | 0.63 ±0.01 | **1.01 ±0.0** | 0.95 ±0.0 | 0.98 ±0.01 | 0.14 ±0.0 |

| MAXIMIZ. | | | | | | |
|---|---|---|---|---|---|---|
| QDDecomp. | **0.61 ±0.0** | **1.0 ±0.0** | 1.01 ±0.0 | 0.9 ±0.01 | **1.01 ±0.0** | 0.03 ±0.0 |
| CondDPP | 0.6 ±0.0 | **1.0 ±0.0** | 1.01 ±0.0 | 0.89 ±0.01 | **1.01 ±0.0** | 0.04 ±0.0 |
| $\varepsilon$-Greedy | **0.61 ±0.0** | **1.0 ±0.0** | 1.01 ±0.0 | 0.9 ±0.01 | **1.01 ±0.0** | 0.03 ±0.0 |
| MarkovDPP | 0.58 ±0.0 | 0.94 ±0.01 | 1.01 ±0.0 | 0.17 ±0.0 | 0.98 ±0.01 | 0.06 ±0.0 |
| **B-DivRec** | **0.61 ±0.0** | 1.0 ±0.0 | 1.01 ±0.0 | 0.95 ±0.0 | **1.01 ±0.0** | 0.12 ±0.0 |

| NO DPP | | | | | | |
|---|---|---|---|---|---|---|
| MMR | 0.6 ±0.0 | **1.0 ±0.0** | 1.01 ±0.0 | **0.96 ±0.0** | **1.01 ±0.0** | 0.08 ±0.0 |

Table 26: Execution time benchmark (runtime in seconds) on SYNTHETIC1500 (4 users, $\tau = 0.5$) with MAXIMIZATION.

| BATCH SIZE $B$ | B=5 | B=50 | B=500 | B=900 |
|---|---|---|---|---|
| MMR | **0.03 ±0.0** | 0.19 ±0.0 | 2.05 ±0.01 | 4.16 ±0.13 |
| **B-DivRec** | 0.07 ±0.0 | **0.08 ±0.0** | **0.21 ±0.0** | **0.32 ±0.0** |
| Ratio MMR/B-DivRec | 0.46 | 2.38 | 9.89 | 12.91 |

# G  ABOUT MOVIELENS AND DIVERSITY

To propose a more quantitative analysis of the performance of B-DivRec and MMR on the Movie-Lens data set, we compute several metrics on all tested real-life data sets in Table 27. For each metric, we also report the ratio of the metric value for our contribution B-DivRec over the metric value for MMR. For instance, using the values from Table 7, we obtain a REL ratio of $\frac{3.48}{3.78} = 0.92$, where $3.78$ is the relevance value achieved by MMR and $3.48$ is the relevance value achieved by B-DivRec. We denote in bold type the cases where the ratio is greater or equal to 1, meaning similar performance or superiority of B-DivRec over MMR.

As a proxy for the implicit feedback bias, we compute the sparsity number, meaning the percentage of observed feedback (that is, usually the number of non-zero values in the user-item rating matrix). The smaller the sparsity number, the greater the bias, as the feedback model is then trained on a smaller number of observed values.

We measure the popularity bias by the Gini coefficient (Braun et al., 2023), denoted Gini in the table, that quantifies inequality in the distributions of item popularity scores. The score is in the range [0,1], where 0 represents no bias, and the larger, the more popularity bias is present.

We also compute metrics regarding the diversity of the user history and data set. History-wise diversity (Hist-Div) represents the global diversity across items in the user history, averaged across users, that is, the collinearity of embeddings of items in the history. History-wise diversity for a given user is DIV$^G$ computed when $S^t = \emptyset$ in Equation 1. Intrinsic diversity (Diversity) is the volume across all items in the data set. We choose to set the volume of an empty set to 0, when the user history is empty.

We also tested new, richer embeddings of movies in MovieLens using the MiniLM 384 model (Wang et al., 2020) instead of the Universal Sentence Encoder model (Cer et al., 2018) to generate embeddings from the movie title and keywords. We name the corresponding data set RicherMovieLens, and we run it with the same parameters as MovieLens in the main text.

Note that Fdataset and Gottlieb have the same rating matrix, but features in Gottlieb are richer (as evidenced by the history-wise diversity value). As mentioned in the paper, in MovieLens, the history-wise diversity is close to 0, meaning that there are at least two embeddings in the history of almost each user which are collinear. This impairs the computation of the score (Equation 2) for B-DivRec, whereas this effect is lessened in MMR, as MMR only relies on the maximum of pairwise diversity scores (Section 2). However, as illustrated by the performance ratios, taking into account diversity over sets incurs a higher diversity in recommendations.

Our assumption is that the poor history-wise diversity metrics on MovieLens are mostly due to collinear item embeddings, where extremely similar movies (like successive entries in a series) have almost the same embedding, collapsing the volume-based diversity metric. A possible solution to mitigate this issue is to consider a "representative" set of items in the user history, instead of the whole history which might contain collinear item embeddings. In Section A, we discuss at length (ridge) leverage scores, which can be used to determine representative points in a set. Representative points are points which are the most "unique", that is, decorrelated from other points.

Table 27: Metrics related to known bias in recommender systems on real-life data sets, and relative performance of B-DivRec compared to MMR.

| DATA SET | MovieLens | PREDICT | Epinions | Gottlieb | Fdataset |
|---|---|---|---|---|---|
| Sparsity (%) | 1.50 | 0.50 | 0.17 | 1.18 | 1.18 |
| Gini | 0.94 | 0.96 | 0.35 | 0.96 | 0.96 |
| Hist-Div | 0.0 | 0.61 | 0.03 | 0.42 | 0.28 |
| Diversity | 0.0 | 0.0 | 0.0 | 0.0 | 0.0 |
| REL ratio | 0.92 | **1.15** | 0.61 | 0.87 | 0.86 |
| $DIV^L$ ratio | **1.09** | **1.48** | **1.00** | **1.19** | **1.82** |
| $DIV^G$ ratio | 0.86 | 0.95 | **1.00** | **1.09** | **1.52** |
| $DIV^+$ ratio | 0.77 | **1.46** | **1.00** | **1.21** | **1.82** |

| DATA SET | Cdataset | DNdataset | LRSSL | RicherMovieLens |
|---|---|---|---|---|
| Sparsity | 1.09 | 0.03 | 0.72 | 1.36 |
| Gini | 0.96 | 1.00 | 0.97 | 0.94 |
| Hist-Div | 0.29 | 0.25 | 0.53 | 0.0 |
| Diversity | 0.0 | 0.0 | 0.0 | 0.0 |
| REL ratio | 0.87 | 0.94 | 0.86 | 0.94 |
| $DIV^L$ ratio | **1.39** | **1.13** | **1.05** | **1.13** |
| $DIV^G$ ratio | **1.54** | **1.13** | **1.01** | 0.98 |
| $DIV^+$ ratio | **1.42** | **1.13** | **1.05** | 0.97 |

Table 28: RicherMovieLens and MovieLens (4 users, $B = 3$, $\tau = 2.5$) with MAXIMIZATION. The results for MovieLens are the same as in Table 7.

| MovieLens ↑ | REL ↑ | $DIV^L$ ↑ | $DIV^G$ ↑ | $DIV^+$ ↑ | TIME ↓ |
|---|---|---|---|---|---|
| MMR | **3.78** $\pm$**0.12** | 0.74 $\pm$0.03 | **0.07** $\pm$**0.01** | **0.77** $\pm$**0.04** | 26.93 $\pm$0.56 |
| **B-DivRec** | 3.48 $\pm$0.13 | **0.81** $\pm$**0.01** | 0.06 $\pm$0.01 | 0.59 $\pm$0.05 | **26.65** $\pm$**0.54** |

| RicherMovieLens ↑ | REL ↑ | $DIV^L$ ↑ | $DIV^G$ ↑ | $DIV^+$ ↑ | TIME ↓ |
|---|---|---|---|---|---|
| MMR | **3.91** $\pm$**0.07** | 0.72 $\pm$0.02 | 0.05 $\pm$0.0 | **0.76** $\pm$**0.02** | **19.00** $\pm$**0.01** |
| **B-DivRec** | 3.68 $\pm$0.06 | **0.81** $\pm$**0.0** | 0.05 $\pm$0.0 | 0.73 $\pm$0.03 | 19.31 $\pm$0.03 |