# OpenReview forum: "Adaptive Quality-Diversity Trade-offs for Large-Scale Batch Recommendation"
_ICLR.cc/2026/Conference — Submitted to ICLR 2026_

### Official Review · Reviewer_sw7v · 2025-10-31

**Soundness:** 2
**Presentation:** 1
**Contribution:** 1
**Rating:** 2
**Confidence:** 3

**Summary:**

B-DivRec aims to maximize both individual- and aggregate-level diversity of an existing model by post-processing while maintaining accuracy.
In this process, users provide feedbacks for their recommendation lists.
B-DivRec modifies DPP and introduces hyperparameters to control trade-off between accuracy and diversity.

**Strengths:**

* B-DivRec targets both individual- and aggregate-level diversity in sequential recommendation.
* Authors provide well-packaged experimental code for reproducibility.

**Weaknesses:**

* The paper is hard to follow.
For instance, the problem definition is scattered across the Notation and Metric sections, and the proposed method is mixed with existing works, making it difficult to clearly distinguish the contributions.
* Problem definition is unclear.
I assume that the problem is to maximize both individual and aggregate diversity while maintaining accuracy by post-processing an existing model for a series of users where they provide feedbacks for each item in the recommendation.
Yet, some details are still unclear such as how users provide their feedback, and how does the feedback model predicting those exist.
* The paper reviews only DPP and MMR for previous works and compares the proposed method with only DPP and its variants.
However, both aggregately and individually diversified recommendations are deeply studied topics so that authors should compare their work with other existing methods.
* The novelty of the proposed method appears insufficient.
As I understand it, the main idea of B-DivRec is:
(1) introducing a trade-off hyperparameter $\lambda$ to balance the contributions of the rating matrix and the similarity matrix in the DPP kernel, and
(2) filtering similar items using a threshold hyperparameter $\alpha$.
Compared to the conventional DPP framework, this approach seems incremental, as it mainly involves adding a few hyperparameters and performing hyperparameter tuning.
Furthermore, the paper does not clearly explain what specific challenges this idea aims to address, nor why introducing these hyperparameters is an effective way to tackle them.
* Proposed metric seems highly sensitive to the threshold $\tau$, which may lead to unfair experimentation.
* Experiments are performed mainly on synthetic datasets rather than real datasets.
* Backbone recommendation models are not clearly specified.
* Performance improvement of the proposed method is marginal.

**Questions:**

* Please refer to weakneses above.
* How does the recommender system receive the user feedback for the recommendation results during the experiments?
Real-world datasets such as MovieLens contain user feedback for only interacted items which are very little, so most recommended items would not be interacted with the user.
Moreover, if duplicate recommendations of items seen in the training dataset are not allowed as usual, conducting such experiments would have been more challenging.

---

> ### Author Response · Authors · 2025-11-20
> **Rebuttal (1/3)**
>
> Thank you for taking the time to review our paper, which targets the tradeoff between both individual ("local") and aggregated ("global") diversity and accuracy in recommendations of batches of items. We address your concerns below:
>
> 1. Experimental setting: **"Yet, some details are still unclear such as how users provide their feedback, and how does the feedback model predicting those exist. [...] How does the recommender system receive the user feedback for the recommendation results during the experiments? Real-world datasets such as MovieLens contain user feedback for only interacted items which are very little, so most recommended items would not be interacted with the user. Moreover, if duplicate recommendations of items seen in the training dataset are not allowed as usual, conducting such experiments would have been more challenging."**
>
> As we do not have access to an online setting where we could directly interrogate users, we had to set up an offline framework on observed ratings for real-life data sets. The details are present in Appendix D, due to the space limitations of the main text. For instance, with the MovieLens data set, we trained a SVD model on the observed movie ratings, which we query during our offline evaluation to obtain an estimated feedback value.
>
> - **" I assume that the problem is to maximize both individual and aggregate diversity while maintaining accuracy by post-processing an existing model for a series of users where they provide feedbacks for each item in the recommendation. [...] Backbone recommendation models are not clearly specified."**
>
> The objective is to study and optimize the tradeoff between accuracy and different levels of diversity (local and global) in recommendations of batches of items. BDivRec, as all other considered baselines, is applied directly on the (whole) library of items and outputs batches of recommended items, hence the importance of tractable computations. Moreover, its formulation exploits the estimated feedback values from a backbone (feedback) model. The strength of our approach is that we can consider any backbone model $q_\theta$ for the feedback (e.g., SVD for MovieLens, heterogeneous graph attention network for PREDICT, trained offline on observed data), as long as it satisfies the assumptions 3.1-3.2. Note that those two assumptions are unrestrictive in practice.
>
> To clarify those concerns, and provided a supplementary page in the camera-ready version of the paper, we will add to the paper a diagram representing our contribution BDivRec, the interactions with the (pseudo) user, the backbone feedback model and the item embeddings.
>
> - **"Proposed metric seems highly sensitive to the threshold $\tau$, which may lead to unfair experimentation."**
>
> Note that the quality (REL) and diversity-specific metrics (DIV$^L$, DIV$^G$) do not depend on a threshold, and can be readily compared across metrics. As regard to the DIV$^+$ metric, although we acknowledge its dependency on a threshold, we believe that this does not significantly bias the evaluation for two reasons:
> - All of the (real-life) data sets in our paper showcase feedback values which are interpretable (MovieLens: number of stars in a movie rating by a user, PREDICT: binary indicator of the success of a clinical trial involving a drug compound and a disease). As such, the definition of a threshold is straightforward and has a meaning: for instance, $\tau=2.5$ in MovieLens means that we favor movies with more than 2 stars, and $\tau=0.5$ means that we favor drug compounds with an estimated probability of success in treating the disease greater than 50%.
> - What matters in assessing the quality and diversity of recommendations is that recommended items are *good* (= have a good chance of satisfying the user) and *diverse* (= are dissimilar to items in the user history or in previous recommendations). As such, the most important point is that the relevance of an item is *high enough*, and the exact (estimated or observed) feedback value is not necessary. This exact feedback value is however captured by the REL metric.
>
> As such, the metric DIV$^+$ tries to capture the requirements of good and diverse items, which is why we proposed this metric in our paper.
>
> 2. Experimental results: **"The paper reviews only DPP and MMR for previous works and compares the proposed method with only DPP and its variants. However, both aggregately and individually diversified recommendations are deeply studied topics so that authors should compare their work with other existing methods."**
>
> Please refer to the rebuttal to Reviewer r1zz on Experimental results, where we tested two new baselines on a synthetic and a real-life data sets. Those two baselines are however not optimizing for the quality-*global* diversity tradeoff, but are only suitable for local diversity. As mentioned in the paper, the closest baseline that we identified is MarkovDPP, which is present in our experiments.

---

> ### Author Response · Authors · 2025-11-20
> **Rebuttal (2/3)**
>
> - **"Experiments are performed mainly on synthetic datasets rather than real datasets."**
>
> Synthetic data sets allow us to test quickly the sensitivity of our method to hyperparameters, and to see readily the impact of the number of items on the performance of the tested methods. We also included two real-life data sets, MovieLens and PREDICT, in our experiments. Furthermore, we now run additional experiments on a new third data set called Epinions (downloadable at https://www.kaggle.com/datasets/masoud3/epinions-trust-network?select=epinions). This data set contains users' ratings (from 1 to 5) on items. We use the same approach as for MovieLens to define the feedback model (see Appendix D). The configuration (apart from the data set) corresponds to the configuration reported in the paper for PREDICT (batch size, hyperparameter values, etc.). We only kept users with at least 50 ratings, and items with at least 100 ratings (due to the time limit on the rebuttal), and used as item embedding matrix the first factor of the SVD (160,417 items in total). We report in the table below the average metrics across 10 iterations.
>
> Epinions | REL | DIV$^L$ | DIV$^G$ | DIV$^+$ | Time (seconds)
> -----|-----|--------|--------|-----|-----
> xQuAD    | **0.038** | 0.001 | 0.0 | 0.0 | 51.942
> MMR    | **0.038** | 0.001 |0.0 | 0.0 | **3.011**
> B-DivRec   |0.023 | 0.001 | 0.0 | 0.0 | 17.155
>
> We observe the same issue as for MovieLens, where the embeddings of items in the user history are collinear, such that the volume of the determinant collapses, and no improvement on the diversity can be obtained. We discuss ways to overcome this issue in our rebuttal to Reviewer r1zz on Experimental results.
>
> Moreover, we also ran experiments on other drug repurposing data sets, namely, Fdataset **[1]**, Cdataset **[2]**, DNdataset **[3]**, LRSSL **[4]** and Gottlieb **[5]**. To obtain the feedback model, we used the same procedure as for PREDICT, as described in Appendix D.
>
> Fdataset | REL | DIV$^L$ | DIV$^G$ | DIV$^+$ | Time (seconds)
> -----|-----|--------|--------|-----|-----
> DeepDPP    | 0.477 | 0.474 | 0.117 | 0.587 | 0.279
> xQuAD    |   0.932 | 0.37 |   0.141 | 0.37 | 0.436
> MMR    | **0.947** | 0.353 | 0.155 | 0.353 | 0.268
> B-DivRec   | 0.814 | **0.644** | **0.235** | **0.644** | **0.191**
>
> Cdataset | REL | DIV$^L$ | DIV$^G$ | DIV$^+$ | Time (seconds)
> -----|-----|--------|--------|-----|-----
> DeepDPP    | 0.355  | 0.478  | 0.156 | 0.323 | 0.37
> xQuAD    | 0.905 | 0.366 | 0.137 |  0.366 | 0.538
> MMR    | **0.914** | 0.363 | 0.142 | 0.363 | 0.352
> B-DivRec   |0.794  |**0.504** | **0.218** | **0.517** | **0.252**
>
> DNdataset | REL | DIV$^L$ | DIV$^G$ | DIV$^+$ | Time (seconds)
> -----|-----|--------|--------|-----|-----
> DeepDPP    | 0.234 | 0.807 |  0.719 | 0.228 |1.579
> xQuAD    |**0.325**  | 0.823 | 0.766 | 0.217 | 1.613
> MMR    | **0.325** | 0.893 |0.864 |  0.194| 1.573
> B-DivRec   | 0.307|**1.009** | **0.977** |**0.254** | **1.365**
>
> LRSSL | REL | DIV$^L$ | DIV$^G$ | DIV$^+$ | Time (seconds)
> -----|-----|--------|--------|-----|-----
> DeepDPP    | 0.393 |**0.898**  |0.529 |  0.642 | 0.481
> xQuAD    | 0.966 | 0.799 | 0.533 | 0.799 | 0.713
> MMR    | **0.972** | 0.818 | 0.554 | 0.818 |  0.527
> B-DivRec   |  0.834 | 0.855 |**0.557**  |**0.855** |**0.359**
>
> Gottlieb | REL | DIV$^L$ | DIV$^G$ | DIV$^+$ | Time (seconds)
> -----|-----|--------|--------|-----|-----
> DeepDPP    |0.504  | 0.642 |0.187 | 0.547 | 0.443
> xQuAD    |  0.933 | 0.645 | 0.363 |0.646  | 0.878
> MMR    | **0.944** | 0.641 | 0.369 | 0.641 | 0.719
> B-DivRec   |  0.822 | **0.764** | **0.404** | **0.776** | **0.435**
>
> **[1]** Gottlieb, et al. Mol. Syst. Biol. 7, 496 (2011).
>
> **[2]** Luo, et al. Bioinformatics 32, 2664–2671 (2016).
>
> **[3]** Martinez, et al. Artif. Intell. Med. 63, 41–49 (2015).
>
> **[4]** Liang, et al. Bioinformatics 33, 1187–1196 (2017).
>
> **[5]** Gao, et al. Front. Pharmacol. 12, 784171 (2022)

---

> ### Author Response · Authors · 2025-11-20
> **Rebuttal (3/3)**
>
> - **"Performance improvement of the proposed method is marginal"**
>
> As illustrated by our newest experiments in the rebuttal, our proposed method BDivRec (1) achieves the quality-(local and global) diversity tradeoff in recommendations, being often performant than MMR, MarkovDPP, xQuAD and Deep DPP, (2) while remaining tractable, even when facing millions of items, contrary to its closest baseline MarkovDPP. Moreover, the proposed method for the adaptive tuning of the level of diversity in recommendations either preserves the quality and diversity of the recommendations, or improves the relevance, which shows that user attrition might be lesser than in the non-adaptive setting.
>
> - **The novelty of the proposed method appears insufficient. As I understand it, the main idea of B-DivRec is: (1) introducing a trade-off hyperparameter $\lambda$ to balance the contributions of the rating matrix and the similarity matrix in the DPP kernel, and (2) filtering similar items using a threshold hyperparameter $\alpha$. Compared to the conventional DPP framework, this approach seems incremental, as it mainly involves adding a few hyperparameters and performing hyperparameter tuning."**
>
> The conventional DPP literature ignores the global diversity problem, as most of the available quality-diversity DPP decompositions aim at optimizing the accuracy-local diversity tradeoff, as discussed in Section 2. As far as our knowledge goes, MarkovDPP (which is one of the tested baselines in our experiments) and our contribution BDivRec are the only methods that try to optimize for the global diversity. The main issue with MarkovDPP, as illustrated in our experiments, is that MarkovDPP is computationally intractable beyond 5,000 items. A key part of BDivRec is the careful combination of several tricks (listed in Appendix B) which allows it to run on data sets with 15 millions of items. We discuss the time complexity of BDivRec in Section 3, on page 5.
>
> As for our other contributions, our paper introduced a unified framework for generic quality-(local and global) diversity DPP decompositions, and the first approach in the literature for automatically tuning the diversity level to the user. This approach can be applied to any DPP in the QDT family, which include traditional quality-diversity decompositions.
>
> Moreover, the adaptive approach for tuning the diversity level goes beyond hyperparameter tuning. The problem we solve is to perform well in terms of quality-diversity tradeoff under *possibly adversarial* conditions (e.g., the user changes their mind and is more interested in some category of items whereas, in the previous recommendation rounds, they might be interested in exploring the set of available items), while hyperparameter tuning assumes at least some stationarity in the user behavior. Then, as in our rebuttal to Reviewer SNoM, framing the problem of tuning the level of diversity to the user as a game allows us to benefit of guarantees on the maximum amount of penalty/user attrition (i.e., the regret defined in Equation 6). Finally, the tuning of the level of diversity must be done "online", meaning while recommending items to the user, whereas hyperparameter tuning is performed on a training data set. The issue is that, especially for new users, such training data set does not exist.
>
> - **"Furthermore, the paper does not clearly explain what specific challenges this idea aims to address, nor why introducing these hyperparameters is an effective way to tackle them."**
>
> As described in Section 2 and Appendix A, introducing a tradeoff parameter to balance accuracy and diversity is a common ingredient of recommender systems, for instance, for MMR, xQuAD or the proposed method in **[1]**. Filtering steps in recommender systems are also common in the literature **[2-3]**. Then, resorting to those techniques to obtain good and diverse recommendations seems intuitive with regards to the literature.
>
> **[1]** Xu and Matsumura. "A Serendipitous Recommendation System Considering User Curiosity." International Conference on Information Integration and Web Intelligence. Cham: Springer Nature Switzerland, 2024.
>
> **[2]** Ibrahim, et al. "Diversified recommendations of cultural activities with personalized determinantal point processes." Second International Workshop on Recommender Systems for Sustainability and Social Good, 2025.
>
> **[3]** Hron, et al. "On component interactions in two-stage recommender systems." Advances in neural information processing systems 34 (2021): 2744-2757.

---

### Official Review · Reviewer_r1zz · 2025-10-31

**Soundness:** 3
**Presentation:** 3
**Contribution:** 2
**Rating:** 4
**Confidence:** 2

**Summary:**

The paper proposes a unified DPP-based framework that cleanly separates quality and diversity with an explicit weight $\lambda$, subsuming prior DPP variants. Building on this, B-DivRec introduces a “fuzzy denuding” step that filters items too close to a user’s history in feature space, enabling scalable global diversity with linear-in-N computation via Nyström and fast MAP/ $\alpha$ -DPP routines. An adaptive $\lambda$ procedure (AdaHedge-style) tunes the quality-diversity balance per user online. Experiments on synthetic datasets up to 15M items, MovieLens, and a drug repurposing benchmark show competitive or superior trade-offs compared to conditional DPP and MMR; notably, B-DivRec is strong on PREDICT while MMR dominates on MovieLens, and adaptive $\lambda$ improves relevance in some settings.

**Strengths:**

- The paper presents a unified DPP-based formulation with an explicit trade-off parameter $\lambda$, providing a clear theoretical foundation that encompasses several existing diversity-aware recommendation methods such as conditional DPP and MMR.
- The proposed B-DivRec approach combines a denuding operation in feature space with Nystrom approximation, achieving linear scalability and enabling large-scale batch recommendation with explicit control over both global and local diversity.
- The adaptive update of $\lambda$ per user is conceptually appealing, allowing personalized control of the quality–diversity balance and offering a promising direction for user-adaptive recommendation systems.

**Weaknesses:**

* Despite its theoretical elegance, the paper does not demonstrate consistent empirical superiority of the proposed method. On the MovieLens benchmark, MMR achieves higher relevance scores than B-DivRec; the explanation (history-vector collinearity) is qualitative and lacks deeper quantitative analysis.
* The overall effectiveness of B-DivRec appears dataset-dependent, strong on PREDICT but weaker on MovieLens, raising questions about robustness and generality across domains with different diversity characteristics.
* The experimental evaluation includes only classical baselines and omits stronger modern re-ranking or diversification methods (e.g., xQuAD, deep DPPs, intent-aware models), so the practical advantage remains unclear.
* The adaptive  $\lambda$ update lacks formal guarantees (e.g., convergence or regret bounds) and is tested only under clean, noise-free feedback, limiting confidence in real-world, noisy environments.
λ update lacks formal guarantees (e.g., convergence or regret bounds) and is tested only under clean, noise-free feedback, limiting confidence in real-world, noisy environments.

**Questions:**

* Could you provide quantitative analysis for why MMR outperforms B-DivRec on MovieLens (e.g., effects of popularity bias, embedding collinearity, or limited intrinsic diversity), and how B-DivRec might be adapted to mitigate these factors?
* Would B-DivRec improve on MovieLens with richer embeddings (e.g., hybrid content–collaborative features) or with per-cluster tuning of $\alpha$ and $\lambda$?
* Can the adaptive $\lambda$ be evaluated under noisy/implicit feedback (clicks, exposure bias) to assess robustness and practicality?

---

> ### Author Response · Authors · 2025-11-20
> **Rebuttal (1/4)**
>
> Thank you for recognizing our unified and scalable personalized quality-diversity framework and the competitive or superior performance of our contribution compared to baselines. We address your concerns below:
>
> 1. Experimental results: **"The paper does not demonstrate consistent empirical superiority of the proposed method. On the MovieLens benchmark, MMR achieves higher relevance scores than B-DivRec; the explanation (history-vector collinearity) is qualitative and lacks deeper quantitative analysis. [...] The overall effectiveness of B-DivRec appears dataset-dependent [...] Could you provide quantitative analysis for why MMR outperforms B-DivRec on MovieLens (e.g., effects of popularity bias, embedding collinearity, or limited intrinsic diversity), and how B-DivRec might be adapted to mitigate these factors?"**
>
> To propose a more quantitative analysis of the performance of B-DivRec, we compute several metrics on all tested real-life data sets, including the newest ones we added to our rebuttal to Reviewer sw7v. For each metric, we also report the ratio of the metric value for our contribution B-DivRec over the metric value for MMR. For instance, using the values from Table 15 in the appendix of the paper, we obtain a REL ratio of $\frac{3.48}{3.78}=0.92$, where $3.78$ is the relevance value achieved by MMR and $3.48$ is the relevance value achieved by B-DivRec. We denote in bold type the cases where the ratio is greater or equal to 1, meaning similar performance or superiority of B-DivRec.
>
> As a proxy for the implicit feedback bias, we compute the sparsity number, meaning the percentage of observed feedback (that is, usually the number of non-zero values in the user-item rating matrix). The smaller the sparsity number, the greater the bias, as the feedback model is then trained on a smaller number of observed values.
>
> We measure the popularity bias by the Gini coefficient (see for instance **[1]**) that quantifies inequality in the distributions of item popularity scores. The score is in the range [0,1], where 0 represents no bias, and the larger, the more popularity bias is present.
>
> We also compute metrics regarding the diversity of the user history and data set. History-wise diversity represents the global diversity across items in the user history, averaged across users, that is, the collinearity of embeddings of items in the history. History-wise diversity for a given user is DIV$^G$ computed when $S^t=\emptyset$ in Equation 1 in the paper. Intrinsic diversity is the volume across all items in the data set. We choose to set the volume of an empty set to $0$ (when the user history is empty).
>
> DATA | MovieLens | PREDICT | Epinions | Gottlieb | Fdataset | Cdataset | DNdataset | LRSSL
> -----|-----|--------|--------|-----|-----|-----|-----|-----
> Implicit feedback bias   | 1.498% | 0.502% | 0.172% |  1.180% | 1.180% |  1.094% | 0.034% | 0.721%
> Popularity bias | 0.937 | 0.956 | 0.345 | 0.959 | 0.959 | 0.962 | 0.999 | 0.971
> History-wise diversity  | 0.0 | 0.607 | 0.034 | 0.421 |  0.276 | 0.293 |0.252  | 0.528
> Intrinsic diversity  | 0.0 | 0.0 | 0.0 | 0.0 | 0.0 | 0.0 | 0.0 | 0.0
> REL ratio   | 0.92 | **1.15** | 0.61 | 0.87 | 0.86 | 0.87 | 0.94 | 0.86
> DIV$^L$ ratio  | **1.09** | **1.48** | **1.00** | **1.19**  | **1.82** | **1.39** |**1.13**  | **1.05**
> DIV$^G$ ratio   | 0.86 | 0.95 | **1.00**  | **1.09** | **1.52** | **1.54** | **1.13** | **1.01**
> DIV$^+$ ratio | 0.77 | **1.46** | **1.00**|**1.21**  | **1.82** |**1.42** | **1.13** | **1.05**
>
> Note that Fdataset and Gottlieb have the same rating matrix, but features in Gottlieb are richer (as evidenced by the history-wise diversity value). As mentioned in the paper, in MovieLens, the history-wise diversity is close to 0, meaning that there are at least two embeddings in the history of almost each user which are collinear. This impairs the computation of the score (Equation 4 in the paper) for B-DivRec, whereas this effect is lessened in MMR, as MMR only relies on the maximum of pairwise diversity scores (see Section 2 of the paper). However, as illustrated by the performance ratios, taking into account diversity over sets incurs a higher diversity in recommendations.
>
> A possible solution to mitigate this issue is to consider a "representative" set of items in the user history, instead of the whole history which might contain collinear item embeddings. In Appendix (Section A), we discuss at length (ridge) leverage scores, which can be used to determine representative points in a set. Representative points are points which are the most "unique", that is, decorrelated from other points.
>
> **[1]** Braun, et al. "Metrics for popularity bias in dynamic recommender systems." arXiv:2310.08455, 2023.

---

> ### Author Response · Authors · 2025-11-20
> **Rebuttal (2/4)**
>
> - **"Would B-DivRec improve on MovieLens with richer embeddings (e.g., hybrid content–collaborative features) or with per-cluster tuning of $\alpha$ and $\lambda$?"**
>
> We ran new experiments on the MovieLens data sets with richer embeddings, using the MiniLM 384 model instead of the Universal Sentence Encoder model to generate embeddings from the movie title and keywords. We name the corresponding data set RicherMovieLens, and we run it for T=3 and only on user "0" due to the time constraints on the rebuttal. All other parameters are left unchanged compared to the paper.
>
> RicherMovieLens | REL | DIV$^L$ | DIV$^G$ | DIV$^+$ | Time (seconds)
> -----|-----|--------|--------|-----|-----
> MMR    | **4.769** | 0.712 | 0.0 | 0.082|**19.257**
> B-DivRec (MAX)   | 4.519 |**0.842** |0.0  |  **0.332** | 19.407
>
> MovieLens | REL | DIV$^L$ | DIV$^G$ | DIV$^+$ | Time (seconds)
> -----|-----|--------|--------|-----|-----
> MMR |  **4.76**| 0.661| 0.0  |0.076 |19.86
> B-DivRec (MAX) |  4.459| **0.788** | 0.0  |**0.309** |**19.795**
>
> DATA | RicherMovieLens on user 0 | MovieLens on user 0
> -----|-----|--------
> Implicit feedback bias   | 1.358% |   1.7%
> Popularity bias  |0.938  | 0.716
> History-wise diversity  |0.0  | 0.0
> Intrinsic diversity  | 0.0 | 0.0
> REL ratio    | 0.95 | 0.94
> DIV$^L$ ratio   | **1.18** |  **1.19**
> DIV$^G$ ratio   | **1.00** | **1.00**
> DIV$^+$ ratio  | **4.05**  | **4.07**
>
> With T=-1 and all the users for MovieLens considered in the paper:
>
> RicherMovieLens | REL | DIV$^L$ | DIV$^G$ | DIV$^+$ | Time (seconds)
> -----|-----|--------|--------|-----|-----
> MMR    | **3.914** | 0.719 | **0.046** |  **0.756** |**18.995**
> B-DivRec (MAX)   | 3.679 | **0.811** | 0.045 |  0.730 | 19.306
>
> In the paper:
>
> MovieLens | REL | DIV$^L$ | DIV$^G$ | DIV$^+$ | Time (seconds)
> -----|-----|--------|--------|-----|-----
> MMR    | **3.78** | 0.74 | **0.07** | **0.77** | 26.93
> B-DivRec (MAX)    | 3.48 | **0.81** | 0.06 | 0.59 | **26.65**
>
> DATA | RicherMovieLens (6 users) | MovieLens (6 users)
> -----|-----|--------
> Implicit feedback bias   | 1.358% |   1.498%
> Popularity bias  |0.938  | 0.937
> History-wise diversity  |0.0  | 0.0
> Intrinsic diversity  | 0.0 | 0.0
> REL ratio    | 0.94 | 0.92
> DIV$^L$ ratio   | **1.13** |  **1.09**
> DIV$^G$ ratio   | 0.98 | 0.86
> DIV$^+$ ratio  | 0.97  | 0.77
>
> Due to the time constraints on the rebuttal, we did not experiment with clustering users or items for MovieLens. As per the previous section, our assumption is that the poor history-wise diversity metrics on this dataset are mostly due to collinear item embeddings, where extremely similar movies (like successive entries in a series) have almost the same embedding, collapsing the volume-based diversity metric.

---

> ### Author Response · Authors · 2025-11-20
> **Rebuttal (3/4)**
>
> - **"The experimental evaluation includes only classical baselines and omits stronger modern re-ranking or diversification methods (e.g., xQuAD, deep DPPs, intent-aware models), so the practical advantage remains unclear."**
>
> We run an experiment with B-DivRec (MAXIMIZATION strategy) and two new baselines, xQuAD and Deep DPP, on SYNTHETIC1000.
>
> For xQuAD (https://dl.acm.org/doi/10.1145/1772690.1772780), we consider as sub-query generation procedure the selection of the items in the history with a cosine similarity higher than $\alpha$. To implement this, we resort to FAISS trees as described for BDivRec. The rationale behind this choice is to be as fair as possible with respect to BDivRec, and in particular to incorporate some information regarding the interbatch/global diversity as defined in our paper. The main difference between xQuAD and BDivRec is that xQuAD only considers the relevance scores of items and similar items in the history in its ranking score.
>
> For DeepDPP (https://web3.arxiv.org/pdf/1811.07245v1), we define the observed subsets needed for learning the likelihood matrix as batches of items of at most B in the user history. Contrary to BDivRec, DeepDPP does not include the estimated feedback values nor the diversity values. If the user history is empty, the likelihood matrix is set to the identity matrix.
>
> We report the average metrics over 10 iterations below:
>
> SYNTHETIC1500 (4 users) | REL | DIV$^L$ | DIV$^G$ | DIV$^+$ | Time (seconds)
> -----|-----|--------|--------|-----|-----
> xQuAD    | **0.645** |  0.003 |  0.0 | 0.003 | 0.757
> DeepDPP    | 0.496 | 0.997 | 0.324 | 0.976 | 0.701
> B-DivRec   | 0.534 | **1.002** |  **0.94** | **1.005** | **0.059**
>
> We ran the same experiments on the PREDICT data set (with the same configuration as in the paper, except that we test for 4 users instead of 9, due to the time constraints on the rebuttal).
>
> PREDICT | REL | DIV$^L$ | DIV$^G$ | DIV$^+$ | Time (seconds)
> -----|-----|--------|--------|-----|-----
> xQuAD    | **0.792** | 0.66 | 0.474 | 0.636 | 0.996
> DeepDPP   | 0.213  | 0.639 | 0.354 | 0.298 | 0.725
> B-DivRec   | 0.679 | **0.864** | **0.517** | **0.804** | **0.501**
>
> The results are somewhat expected, as xQuAD do not take into account explicitly the similarity scores between items and the corresponding algorithm has almost the same structure as MMR. DeepDPP leverages the power of DPPs to perform well (local) diversity-wise, but is worst at relevance. BDivRec clearly improves upon all baselines either in terms of diversity (xQuAD) or relevance (Deep DPP), highlighting the quality-diversity tradeoff we aimed for. However, note that xQuAD and DeepDPP were not developed for the optimization of the quality-global diversity tradeoff.
>
> 2. Theoretical results: **"The adaptive $\lambda$ update lacks formal guarantees (e.g., convergence or regret bounds) and is tested only under clean, noise-free feedback, limiting confidence in real-world, noisy environments."**
>
> Please refer to the rebuttal to Reviewer SNoM on Theoretical results for the derivation of an upper bound on the regret defined in Equation (6) of the paper, and a discussion on the extension of our approach to noisy environments.

---

> ### Author Response · Authors · 2025-11-20
> **Rebuttal (4/4)**
>
> - **"Can the adaptive $\lambda$ be evaluated under noisy/implicit feedback (clicks, exposure bias) to assess robustness and practicality?"**
>
> We consider the setting where the recommender system only has access to the outcomes of clinical trials (3: successful, 2: not tested, 1: unsuccessful) from the PREDICT data set, or to the movie ratings from 1 (not seen), 2 (rated 1 star) to 6 (five stars) from the MovieLens data set. The choice of the outcome values is determined by our assumption of positive feedback values. In both PREDICT and MovieLens, similarly to what we described in the paper, we observe an improvement of the relevance (to the price of some of the diversity values) when using the adaptive tuning procedure, and the procedure is able to almost retrieve the best a posteriori value $\lambda^\star$ from the starting value $\lambda^0=0.5$. In particular, in MovieLens, the algorithm almost always returns top-rated movies in the successive 233 batches of 3 recommended items for user 0.
>
> PREDICT | $\lambda_\text{final}$ | $\lambda^\star$ | REL | DIV$^L$ | DIV$^G$ | DIV$^+$ | Time (seconds)
> -----|-----|-----|-----|--------|--------|-----|-----
> B-DivRec (adaptive) |  0.974 |  0.993| 2.412 | 0.541 | 0.207 |  0.541| 7.705
> B-DivRec   | $-$ |$-$ |  2.28| 0.83  | 0.33 |0.83 | 1.355
>
> MovieLens | $\lambda_\text{final}$ | $\lambda^\star$ | REL | DIV$^L$ | DIV$^G$ | DIV$^+$ | Time (seconds)
> -----|-----|-----|-----|--------|--------|-----|-----
> B-DivRec (adaptive) |0.997|  0.997 | 5.996 | 0.944|0.02 |  0.944|57.563
> B-DivRec   | -- | -- | 3.57  | 0.944  |0.021  |0.944 | 2.716
>
> We also set up a synthetic environment with 750 items defined similarly to our SYNTHETIC* data sets in the paper, except that the recommender system no longer receives the exact feedback score $q_\theta(\phi_i, \textbf{h}^t)$ for item $i$ and user $\textbf{h}^t$, but a binary outcome in {1,2} determined by a Bernouilli law of probability $p = \min(1, \max(0, q_\theta(\phi_i, \textbf{h}^t)))$. Note that 1 is drawn with probability $1-p$, and 2 with probability $p$. This setting simulates the case where we only receive clicks, and no longer probabilities of clicks. We report below the results compared to B-DivRec applied to the same data set without adaptive tuning of $\lambda$, starting from $\lambda⁰=0.5$. As the relevance is already close to the maximum (2) in this case, there is no observable improvement when running the adaptive procedure.
>
> SYNTHETIC | $\lambda_\text{final}$ | $\lambda^\star$ | REL | DIV$^L$ | DIV$^G$ | DIV$^+$ | Time (seconds)
> -----|-----|-----|-----|--------|--------|-----|-----
> B-DivRec (adaptive) |0.727 |  0.5 | 1.996 |1.012 | 0.164 |  0.068 | 1.504
> B-DivRec   | $-$ | $-$ | 1.996  | 1.012 | 0.164 | 0.068 | 0.061
>
> These experiments validate our findings in the paper even when we only have access to observable feedback (and not to the estimated feedback values by a proxy model). The adaptive procedure allows us to improve on the relevance, when possible. When the relevance cannot be improved further, the relevance-wise and diversity-wise performance remains the same as in the non-adaptive setting.

---

### Official Review · Reviewer_SNoM · 2025-11-01

**Soundness:** 2
**Presentation:** 2
**Contribution:** 2
**Rating:** 4
**Confidence:** 3

**Summary:**

This paper investigates a scalable approach that considers the relevance-diversity trade-off in recommender systems.
Although the proposed framework is simple, the authors provide a detailed discussion of the implementation (especially in the appendices) to account for practical scalability.
While it lacks theoretical contributions such as regret bounds, the paper's rich discussion on practical aspects makes it valuable to the recommender systems community.
On the other hand, the numerical experiments on execution time are limited to small-scale settings (e.g., $B=3$), and the method does not appear to be faster than existing methods (Tables 10-14).
Therefore, the superiority of the proposed method is not sufficiently demonstrated.

**Strengths:**

1. This paper addresses a highly complex yet practical problem: incorporating diversity in a setting where the item set/batch is recommended sequentially to a user.
2. This paper provides comprehensive discussions covering both theory and implementation.

**Weaknesses:**

1. The experimental results lack persuasiveness. Specifically, the paucity of comparisons regarding execution time with existing methods undermines the paper's main claim of scalability.
2. Although the paper uses theoretical notation, its theoretical contribution is limited. For example, it mentions regret but does not discuss an algorithmic regret bound or similar rigorous analysis.
3. The assumption of noiseless feedback (Assumption 3.4) appears highly unrealistic. In the context of recommender systems, there are few practical scenarios where the expected reward can be directly observed.
4. In my opinion, addressing the relevance-diversity trade-off is a means to an end, and the authors should have prioritized a proper problem formulation. Framing the problem using a model like rotting bandits [a] might have enabled a more robust discussion of concepts such as regret.


[a] Rotting bandits, N Levine, K Crammer, S Mannor - Advances in neural information processing systems, 2017

**Questions:**

1. What were the results of the numerical experiments on execution time when varying the batch size?
2. How is the proposed method intended to be executed in scenarios where the expected reward cannot be observed, such as with binary feedback (e.g., clicks)?

---

> ### Author Response · Authors · 2025-11-20
> **Rebuttal (1/2)**
>
> We thank the reviewer for acknowledging the importance of the diversity problem and our detailed and rich discussion of the implementation for practical scalability. We will address the following concerns:
>
> 1. Experimental setting: **"the numerical experiments on execution time are limited to small-scale settings (e.g., B=3) [...] paucity of comparisons regarding execution time with existing methods [...] The method does not appear to be faster than existing methods (Tables 10-14) [...] What were the results of the numerical experiments on execution time when varying the batch size?"**
>
> Note that we considered small batch sizes (B=3, B=5 for SYNTHETIC15M) as ultimately in real-life applications, the user might only be willing to grade up to 5 items at a time. However, we run an experiment with B-DivRec and MMR combined with the MAXIMIZATION strategy on SYNTHETIC1500 with varying batch sizes. We report the average execution time in seconds over 10 iterations below:
>
> ALGO / B | 5 | 50 | 500 | 900 |
> -----|-----|--------|--------|-----
> MMR    | **0.032** | 0.188 | 2.047 | 4.156
> B-DivRec   | 0.069 | **0.079** | **0.207**| **0.322**
>
> Albeit MMR is slightly faster for smaller values of batches, it becomes x12 slower as the batch size increases.
>
> 2. Theoretical setting: **"The assumption of noiseless feedback (Assumption 3.4) appears highly unrealistic. Framing the problem using a model like [rotting bandits](https://papers.nips.cc/paper_files/paper/2017/hash/97d98119037c5b8a9663cb21fb8ebf47-Abstract.html) might have enabled a more robust discussion of concepts such as regret. [...] How is the proposed method intended to be executed in scenarios where the expected reward cannot be observed, such as with binary feedback (e.g., clicks)?"**
>
> The current paper introduces a quality-diversity framework which enables the intuitive (adaptive) tuning of the level of diversity in recommendation. We highlighted this assumption as it is frequently ignored by other recent papers **[1-3]**, but tackling the associated problem in addition to the quality-diversity tradeoff would constitute another contribution by itself. We view dealing with noisy feedback and optimizing for (global) diversity as both important but *independent* problems.
>
> As mentioned in our discussion, overcoming this assumption is an interesting future venue of research, even if the adaptive tuning approach (and the related regret bound, see below) does not need noiseless feedback, and could be applied directly on observed feedback $\hat{y}$. We would assume that applying straightforwardly BDivRec with noisy feedback might have bad results, due to the distribution shift induced by using empirical feedback values instead of the true ones. A first approach to improve upon our framework could implement the optimism principle as in multi-armed bandits and building confidence intervals on the expected feedback on items in the likelihood matrix of a DPP from the QDT family (e.g., replacing the feedback value by a UCB), or leveraging indeed the literature on rotting bandits.
>
> In our rebuttal to Reviewer r1zz, we show that the adaptive procedure described in our paper can still be successfully applied in the case where the recommender system only has access to previously observed feedback.
>
> **[1]** Xu and Matsumura. "A Serendipitous Recommendation System Considering User Curiosity." International Conference on Information Integration and Web Intelligence. Cham: Springer Nature Switzerland, 2024.
>
> **[2]** Fu, Zhe, et al. "A deep learning model for cross-domain serendipity recommendations." ACM Transactions on Recommender Systems 3.3 (2025): 1-21.
>
> **[3]** Li, Fan, et al. "Contextual distillation model for diversified recommendation." Proceedings of the 30th ACM SIGKDD Conference on Knowledge Discovery and Data Mining. 2024.

---

> ### Author Response · Authors · 2025-11-20
> **Rebuttal (2/2)**
>
> 3. Theoretical results: **"Theoretical contribution is limited [...] discuss an algorithmic regret bound or similar rigorous analysis"**
>
> We derive an upper bound on the regret in Equation (6) as follows. Remember that we applied AdaHedge with the following gain function at time $t$ (Line 8 of Algorithm 1), using the computations made in Appendix C for the gradient
> $$\forall \lambda \in [0,1], g_t(\lambda) = (1-2\lambda)\underbrace{\left( -4\log(\text{vol}(f(k,S^t,H^t)))+4\sum_{i \leq B} \log y^t_i\right)}_{= C_t}$$
>
> Note that the gradient $C_t$ is constant with respect to $\lambda$, and that the gain function is linear (and then, concave) in $\lambda$. Applying successively the gradient trick, Theorem 8 and Corollary 17 from the AdaHedge paper (https://arxiv.org/pdf/1301.0534) to the convex loss function $-g_t$ leads to the following regret upper bound
>
> $$ R^\text{adapt}(T;\textbf{h}) \leq \max_{\lambda \in [0,1]} \sum_{t \leq T} g_t(\lambda)-g_t(\lambda^t) \leq 2\delta_T\sqrt{T\log(2)}+16\delta_T(2+\log(2)/3)$$
>
> where $\delta_T = 8 \max_{t \leq T} \log \frac{M_t^B}{a_t}$, where $a_t = \text{vol}(f(k,S^t,H^t)) \in (0,1]$, and $M_t = \max_{i \leq B} y^t_i \geq 1$ for all cases in this paper. Our approach has a regret upper bound in $\mathcal{O}(\sqrt{T})$, which is the best in the related literature on online learning. We will add this proof to the paper.

---

### Author Response · Authors · 2025-11-20
**Global rebuttal**

We thank all reviewers for taking the time to read and comment on our paper. Our paper aimed at introducing a clear and unified framework for the quality-diversity tradeoff in the DPP literature, a tractable algorithm for optimizing the global diversity over millions of items, and the first adaptive approach in the literature to tune the level of diversity in recommendations to the user. Reviewers acknowledged the importance of the diversity problem and of our discussion on the practical implementation (Reviewer SNoM), our unified and scalable personalized quality-diversity framework and the competitive/superior performance of our contribution compared to baselines (Reviewer r1zz), and the reproducibility of our approach (Reviewer sw7v).

We addressed the reviewers' concerns as follows:

- Experimental results: We ran additional experiments to address the question of the runtime depending on the batch size (Reviewer SNoM), the improvement of our contribution BDivRec over the state-of-the-art (Reviewers r1zz and sw7v) and on other real-life data sets (Reviewer sw7v), and the adaptive procedure under a noisy environment (Reviewer r1zz). Moreover, we improved our implementation by using a more efficient data storage format (.npy) and improving the search for neighbors in BDivRec, accounting for a smaller runtime overall and solving the issue raised by Reviewer SNoM.

- Theoretical results: We discussed our reasons behind the assumption of noiseless feedback (Reviewers SNoM and r1zz). We derived an upper bound on the regret defined in Equation 6 in the adaptive quality-diversity tuning procedure (Reviewers SNoM and r1zz).

We included all those changes in the revision of the paper (changes are in blue font). More precisely:
- We merged sections Notation and Metrics into Problem setting.
- We gave the upper bound on regret, and the corresponding proof in Appendix.
- We added all new real-life data sets and baselines to the experimental section, pushing previous experiments to Appendix E due to the space limitation on the paper. The corresponding full numerical results are now in Appendix D.
- We added experiments on noisy feedback with the adaptive procedure in the experimental section.
- We included a diagram of the setting of our paper in Appendix A.
- We added experiments on varying batch sizes in Appendix F.
- We included our discussion and experiments on MovieLens in Appendix G.

---

> ### Author Response · Authors · 2025-11-28
>
> Dear all,
>
> Based on your comments, we submitted individual rebuttals and updated a revision of our paper, where changes are shown in blue font. The extent of the changes is described in the global rebuttal comment. As the deadline draws nearer, we would like to hear your replies concerning our rebuttals, possible other concerns or whether the questions raised in your review were addressed.
>
> We thank you again for your work in reviewing our paper.

---

### Meta-Review · Area_Chair_dtGo · 2025-12-22

**Summary:**

After reviewing the reviewer's concerns, there is agreement that the paper tries to address an important and practically relevant problem: balancing relevance and diversity in large-scale batch recommendation. However, several concerns consistently limit this paper for acceptance. First, reviewers question the strength and generality of the empirical evidence. While the method performs well on some synthetic and selected real-world datasets, results are dataset-dependent, with weaker performance on benchmarks such as MovieLens, and the claimed superiority over baselines is not consistently demonstrated. Multiple reviewers note that comparisons are largely restricted to classical baselines (e.g., MMR, DPP variants), leaving uncertainty about competitiveness against stronger or more modern diversification and re-ranking methods.

Although the paper claims backbone-model agnosticism and suggests that any feedback model $q_\theta$ could be used, in practice the experiments depend on outdated backbone models and datasets (e.g., SVD on MovieLens). Both the model and the benchmark are more than a decade old and no longer reflect the capabilities or challenges of modern recommender systems. As a consequence, the empirical validation does not convincingly demonstrate the method’s relevance or effectiveness in contemporary recommendation pipelines, nor does it provide a sufficiently strong foundation for the paper’s theoretical or practical claims. As a result, this paper does not reach the acceptance standard.

**Reviewer Concerns:**

I believe that the reviewer's concerns from both theoretical and experimental are still not addressed by the rebuttal.

**Reviewer Scores:**

I think most reviewers would not change their scores. The additional experimental results still follow the same issue to the original results, which are questioned by the reviewers.  Also, the rebuttal for the concern of outdated backbone model does not tackle the core part of the concern.  The reply does not convince from both 1) why not use modern backbones, and 2) why only applying the modern models for recommendation would suffer this diversity issue.

---

### Decision · Program_Chairs · 2026-01-26

Reject